

# Efficient pan-European river flood hazard modelling through a combination of statistical and physical models

Dominik Paprotny, Oswaldo Morales-Nápoles, Sebastiaan N. Jonkman

Department of Hydraulic Engineering, Faculty of Civil Engineering and Geosciences, Delft University of Technology,
Stevinweg 1, 2628 CN Delft, The Netherlands.

*Correspondence to*: D. Paprotny (d.paprotny@tudelft.nl)

**Abstract.** Flood hazard is being analysed with ever-more complex models on national, continental and global scales. In this paper we investigate an alternative, simplified approach, which combines statistical and physical models in order to carry out flood mapping for Europe. Estimates of extreme river discharges made using a Bayesian Network-based model from a previous study are employed instead of rainfall-runoff models. Those data provide flood scenarios for simulation of water flow in European rivers with a catchment area above 100 km². The simulations are performed using a one-dimensional 'steady-state' hydraulic model and the results are post-processed using geographical information system (GIS) software in order to derive flood zones. This approach is validated by comparison with Joint Research Centre's (JRC) pan-European map and five local flood studies from different countries. Overall, both our and JRC's maps have similar performance in recreating flood zones of local maps. The simplified approach achieved similar level of accuracy, while substantially reducing the computational time. The paper also presents the summarized results from the flood hazard maps, including future projections. We find that relatively small changes in flood hazard are observed (increase of flood zones area by 2–4%). However, when current flood protection standards are taken into account, there is a sharp increase in flood-prone area in the future (28–38% for a 1000year return period). This is because in many parts of Europe river discharge with the same return period is projected to increase in the future, thus making the protection standards insufficient.

**Keywords.** hydrology; climate change; return periods; flood risk; model comparison

## 1 Introduction

River floods are one of the most common natural hazards in Europe. This fact results in a large number of local and national studies creating flood hazard and risk maps. On the one hand they provide high-resolution information for flood risk management, on the other they usually do not include climate change scenarios. Also, the socio-economic environment changes rapidly, while every few years a new generation of climate projections is produced. Furthermore, the EU Flood Directive requires revisions of flood maps every six years (European Union 2007). Yet, costs of detailed studies are high – for England (2005–2013) the bill was £7 million (approx. €10 million), not including the necessary surveys and data



collection, which amounted to more than £20 million (Environment Agency 2016). The scope and extent of the studies varies across Europe, as does the level of dissemination – few countries make the geospatial data underlying the flood maps easily available. Due to methodological differences, the comparability of the maps is limited, and consequently the possibility of aggregating them and drawing European-wide conclusions is also hampered. Outside the continent, local flood maps are often not present at all.

To produce spatially-consistent maps over large areas, several studies on European and global river flood hazard studies were commissioned. In Europe a series of studies was recently made (Alfieri et al. 2014, 2015a, 2015b) using the Lisflood model (van der Knijff et al. 2010) to derive 100-m resolution maps for the continent. The same model is also used in the European Flood Alert System EFAS (Thielen et al. 2009) as well as its global extension, Global Flood Alert System GloFAS (Dottori et al. 2016). On a global scale, recent river floods studies include GLOFRIS (Winsemius et al. 2013, Ward et al. 2013), SSBN (Sampson et al. 2015) and analyses based on CaMa-Flood model (Pappenberger et al. 2012, Hirabayashi et al. 2013). The resolution of the resulting maps ranges from 3" to 30", or approximately 90–900 m on the equator. Methodologies employed in those studies vary, but mostly start with coarsely-gridded simulation of river flows based on meteorological and land surface data. Flood volumes calculated at 0.25–0.5° resolution are typically downscaled and redistributed over finer grid cells to generate flood extents. In studies based on Lisflood model, a 2D hydrodynamic simulation was performed. However, validation of the models' accuracy has been limited over Europe; only Alfieri et al. (2014) and Sampson et al. (2015) directly compare flood zones they computed with local high-resolution studies. The practical use of the maps is also limited by rather small availability of the underlying data; they are mostly available as online visualizations or through direct contact with the authors. Additionally, the common assumption of those maps is that there are no flood defences in place, thus constituting a "worst-case" scenario (Jonkman 2013, Ward et al. 2015). On the other hand, most of them do—or can—incorporate climate change and socio-economic developments needed to analyse flood frequency over time.

Calculating flood hazard for the whole continent or the globe in one go is computationally demanding, too. Alfieri et al. (2014) mentions using a 60-processor cluster to perform a two-dimensional (2D) simulation of flood zones at 100 m resolution for one scenario only. Sampson et al. (2015) indicated that a similar calculation (3" grid, 2D model) would take 3 months on a single processor core for an average 10° x 10° grid box, which is roughly the geographical extent of metropolitan France. Using a 200-core cluster, the time is reduced to less than a day. Still, the question remains if using complex models is necessary given the quality and resolution of the input data. Bates and De Roo (2000) compared, for a case study in the United Kingdom, output from three different model types with extents of an actual flood. They found that at 100 m resolution a 2D dynamic model performed almost identically to a one-dimensional (1D) steady state, and did improved only slightly compared with extrapolating water levels from observations over the digital elevation model (known as "planar approach"). In another case study in Germany, Apel et al. (2009) found small influence of model choice (water level interpolation, 1D/2D model, 2D model) on the results of a flood risk analysis. Sampson et al. (2015) replaced hydrological modelling of river discharges with a statistical method, the regional frequency analysis (RFA). Applying the



same hydraulic model as in Alfieri et al. (2014) to calculate flood extents, the researchers achieved a better fit to high-resolution flood maps of Thames and Severn river basins than the earlier study. Similar comparison for the two areas, of four global flood models, was presented by Dottori et al. (2016). The results are not conclusive as to which modelling approach gives the best results.

In light of the above, it is not surprising that simpler approaches are still used for flood research. For the CFFlood dataset (Mokrech et al. 2015), for instance, river flood extents were derived by using the planar approach based on water levels computed in Lisflood simulations from Feyen et al. (2012), albeit no validation was presented for either study. As mentioned before, Sampson et al. (2015) utilized a regional frequency analysis of river discharges that was presented by Smith et al. (2015). This study found that river discharges can be estimated by clustering gauge stations based on climate

type, catchment area and annual rainfall. At any location, the discharge could be modelled through similarity of catchment parameters to those clusters. Paprotny and Morales Nápoles (2015, 2016a) employed Bayesian Networks to estimate extreme river discharges in Europe using seven geographical characteristics of catchments. The results have shown that similar accuracy to pan-European studies using hydrologic models could be achieved. Finally, for the lack of better solutions, flood defences were omitted altogether in almost all studies. Occasionally, an assumption that more valuable areas are better

protected was used to compile databases of flood protection standards (Mokrech et al. 2015, Scussolini et al. 2016).

The ultimate aim of the research presented in that paper was to construct flood hazard maps for Europe under present and future climate. This paper builds on the results of Paprotny and Morales Nápoles (2016a). In the aforementioned manuscript, it was explained how extreme river discharges can be derived for the whole continent using only a statistical model, while this paper extends the research by calculating river flood extents in the same area. A relatively simple

combination of one-dimensional hydraulic simulation of water levels and GIS-based planar approach to drawing flood zones is utilized here. Emphasis is put on analysing the accuracy of the results in terms of match with local high-resolution flood maps. This is put in context of the performance of more advanced models in the same areas. Additionally, the aggregate results of the analysis are presented, showing flooded areas at various return periods, the expected changes in the level of hazard due to climate change, and the influence of flood defence standards on the modelling outcomes

It should be noted that the work presented here was a part of a larger effort to create pan-European meteorological and hydrological hazard maps within "Risk analysis of infrastructure networks in response to extreme weather" (RAIN) project. This fact had certain influence on design choices made for the study, such as the extent of the domain, source of input data or representation of the results This was done in order to synchronize the various hazard maps produced within the project (Groenemeijer et al. 2016).



## 2 Materials and methods

### 2.1 Domain and overview of the methodology

The analysis presented here was performed over a domain covering most of the European continent, the same as presented in Paprotny and Morales Nápoles (2016a). This domain excludes most of Russia, Ukraine and Belarus, as well as some outlying island territories, but adds Cyprus, as it is a member of the European Union. In this area there are around 2 million km of rivers in more than 830,000 catchments, according to the CCM River and Catchment Database v2.1, or CCM2 (Vogt et al. 2007, de Jager and Vogt 2010). Despite discharge estimates being available from the aforementioned study for all those rivers, the smallest rivers were removed, as those are generally affected by flash floods. A threshold of 100 km$^2$ upstream area was chosen, which reduces the domain to 155,664 river sections (19% of the total), while retaining 26% of river length (498,420 km). That is still more than double of the 188,300 km of rivers analysed in Alfieri et al. (2014). Global studies mostly used higher thresholds: 5000 km$^2$ in Dottori et al. (2016), which would have reduced our domain to 56,000 km (3%), or Strahler number at least 6 in Winsemius et al. (2013), which would have had almost the same effect. The scope of the paper are river floods, therefore influence of tides and storm surges is not included. Also, flash floods in very small catchments (below 100 km$^2$), which occur over a short period of time, are not covered here.

In this domain flood extents were calculated using the procedure presented in Fig. 1. Firstly, river discharges estimates from the Bayesian Network-based model (I) are collected, as described in section 2.2. Together with data on the river network and terrain (II) they serve as input data for a one-dimensional simulation of water levels using SOBEK model (III). After the water levels (IV) have been calculated as per section 2.3, they are transferred to GIS software (V). Flood zones (VI) are then delimited utilizing the planar approach (section 2.4). The model in SOBEK is then calibrated (VII) based on the comparison with a set of reference maps (VIII), both local high-resolution studies and the Joint Research Centre's (JRC) map (section 2.5). Notice that the calibration step could indicate new runs of the SOBEK model. After the calibration is complete, the resulting flood extents are validated (IX) with additional reference maps and contrasted with the outcomes of other studies (X), which is presented in section 3.1. Finally, flood extents are calculated both for the "reference" period (1971–2000) and climate change scenarios.

### 2.2 River discharge scenarios

In the approach chosen for this study, only the peak discharge value is used in the hydraulic model, rather than flood volumes or time-series of discharges. The "steady-state" simulation calculates the equilibrium water level, there time factor is excluded (see section 2.3). The Bayesian Network-based (BN) model provides estimates of annual maxima of daily river discharge (Paprotny and Morales Nápoles 2016a). By applying extreme value analysis, discharge with different return period is obtained. The calculation was made in the aforementioned paper for three time periods 1971–2000, 2021–2050 and 2071–2100. The first one is the historical "reference" period, used to calibrate and validate the method's performance. The other two represent climate change scenarios, of which two were used for each of the time periods, namely RCP 4.5 and RCP 8.5.



Those "representative concentration pathways" indicate changes in future physical and socio-economic environment that would cause, by 2100, increase in radiative forcing by 4.5 or 8.5 W/m$^2$ (Moss et al. 2010). The BN model is a statistical method that explores the dependencies between the different geographical properties of European catchments, such as size, climatology, terrain and land use. Between the aforementioned time periods all but two variables remain constant. Those

remaining two variables are information from climate models: annual maximum of daily amount of precipitation and snowmelt and the runoff coefficient (annual maximum of total runoff divided by the previously mentioned variable). This allows the method to provide discharge estimates for any climate scenario and time period based on output from climate models (Fig. 2). Here, results from one of the high-resolution regional models operated within EURO-CORDEX framework was used (see Paprotny and Morales Nápoles 2016a for details).

Yet, some additional work was necessary to use the extreme discharge estimates in the hydrodynamic simulation. All large-scale flood assessments face the problem of missing channel geometry data. Most of the time, the problem is solved by using the assumption that the satellite-derived digital elevation model represents the surface water at normal conditions. Thus, only the flow above the surface under "normal conditions" is considered. This baseline flow is therefore subtracted from the peak discharge estimates. It could be the mean annual discharge (Alfieri et al. 2014, Dottori et al. 2016) or the

bankfull discharge, which is assumed to be equal to a 2-year return period (Ward et al. 2013, Sampson et al. 2015). Here we used the former approach, as it gave slightly better results when comparing the flood extents with the reference maps than the other approach. To estimate mean discharge, the BN model was modified by replacing the two variables representing the extreme meteorological events, namely annual maximum of daily precipitation combined with snowmelt and extreme runoff coefficient (annual maximum of total runoff divided by maximum of precipitation and snowmelt), with their equivalents for

average climatology. Therefore, mean annual precipitation and average runoff coefficient (mean annual total runoff divided by mean annual precipitation) are considered. The BN was quantified for 1841 catchments using the same sources of data as before, and contrasted with observations from gauge stations (Fig. 3). The coefficient of determination ($R^2$) is 0.93, which is the same value reported in Rojas et al. (2011) for a hydrological model of Europe without bias-correction of climate data. For specific river discharge, i.e. runoff divided by the respective catchment areas, the $R^2$ is 0.60. The Nash-Sutcliffe

efficiency ($I_{NSE}$), which measures of bias, stands at 0.85. This is better than -0.39 reported in Rojas et al. (2011), though only for the model that was not statistically corrected for bias; the bias-corrected model had a very high $I_{NSE}$ of 0.99.

**2.3 River water level modelling**

Calculation of water levels was performed using SOBEK v2.13 hydrodynamic model (Deltares 2016). As noted in the introduction, the one-dimensional (1D) module was chosen, as it is computationally significantly less demanding than a two-

dimensional (2D) model. One-dimensional flow is described by de Saint-Venant's continuity equation (eq. 1) and momentum equation (eq. 2). In case of momentum equation, the four components describe inertia, convection, water level and bed friction, respectively (Deltares 2016):



$$\frac{\partial Q}{\partial x} + \frac{\partial A}{\partial t} - q = 0 \tag{1}$$

$$\frac{\partial Q}{\partial t} + \frac{\partial}{\partial x}\left(\frac{Q^2}{A}\right) + gA\frac{\partial h}{\partial x} + \frac{gQ|Q|}{C^2RA} = 0 \tag{2}$$

where $Q$ – discharge (m³/s); $x$ – distance (m); $A$ – wetted area (m²); $t$ – time (s); $q$ – lateral discharge per unit length (m²/s) ; $g$ – gravity acceleration (m/s²); $h$ – water level above reference level (m); $C$ – Chézy coefficient (m$^{1/2}$/s); $R$ – hydraulic radius

(m). The momentum equation can be expanded with two more elements (wind friction and extra resistance), but both were not included in this calculation. Also, a "steady-state" calculation was performed, i.e. the model iteratively performs the simulation until an "equilibrium" state of water level for a given discharge amount is found. This means that discharge is assumed non-variable in time, and as a result the inertia term in eq. 2 equals zero. This approach conserves time compared to an "unsteady" calculation, where water levels are calculated for each defined time step. The hydraulic simulation was

prepared utilizing six inputs: river network geometry; river cross-sections; calculation points; upstream and downstream boundaries; lateral discharge; model parameters.

The geometry of the river network was obtained from the linear representation of the rivers in the CCM2 dataset. As noted in section 2.1, river sections with catchment area of at least 100 km² were selected. The network was divided into seven sub-simulations based on the regional split of the original CCM2 dataset (Fig. 4). The resolution of the geometry is

about 100 m. Cross-sections of the rivers were derived from the EU-DEM elevation model (DHI-GRAS 2014) at 100 m resolution. They vary in length depending on the characteristics of the topography (elevation differences), so that the maximum extent of the floodplain is captured. The density of the cross-section along the rivers also varies. CCM2 dataset splits rivers into segment whenever two rivers merge; number of cross-sections per segment depends on its length. On average, the cross-sections are 2.1 km apart. Due to the low resolution of the DEM two assumptions had to be made. Firstly,

that the DEM represents the average water level in the river, as discussed in the previous section. Secondly, that no flood defences or other discharge-control structures are present, unless dikes are large enough to be captured by the DEM. The latter assumption is featured in all continental and global studies, and sometimes even in national studies, such as the flood assessment for England. The aspect of flood protection was dealt with outside the hydraulic computation itself (see section 2.4).

Another input element, calculation points, are locations along the digitised river network where the 1D model computes the water levels. A 1D model represents the rivers and channels as a linear object, therefore allowing movements of water along a single dimension. The dimensions of the river bed and floodplain are defined on the cross-sections. The method utilizes de Saint-Venant equations for continuity and momentum to calculate discharges in a longitudinal profile at calculation points. As another computational-time-conserving simplification, the lumped conveyance approach was used

rather than vertically-segmented approach. This means that it is assumed that velocity is uniform along the profile, as opposed to allowing different velocities in each defined vertical segment. Similarly to cross-sections, calculation points vary



in density and were defined is such a manner that they are located between the cross-sections. Their total number is slightly higher, so that the average distance between them is 2 km.

Water enters and exits the model of the river network at boundaries. Because a threshold of 100 km² catchment area is used, almost all upstream boundaries are located somewhere along the rivers, therefore discharge values were drawn from the BN estimates for that particular location. In rare cases where the source river section already has a catchment bigger than the threshold, the value of discharge was taken from the BN estimate made for that catchment. As noted earlier, average discharge was subtracted from the extreme discharge value for the purpose of the calculation. The downstream boundary is defined differently, as a constant water level, which recreates the river entering the sea. (The only exceptions are two rivers draining to lake Prespa in the southern Balkans). The boundary was defined as zero water level, representing the mean sea level, unless the DEM indicated a value lower than zero. This could be due to river moving through a depression, bias in the DEM or the difference between the mean sea level and modelled geoid underpinning the DEM.

The discharge changes between the upstream and downstream boundaries as it incorporates additional catchment area, therefore more discharge had to be added along the river network. Lateral discharge nodes are used here to enter or withdraw water from the model at locations different than the boundaries. This also is necessary to properly represent the discharge scenarios in the network. At an intersection of two rivers, the water flow in both rivers is summated and continues downstream. However, extreme discharges with e.g. 100-year return period do not necessarily happen at the same time. Hence, the 100-year discharge in the river below the intersection will be typically lower than the sum of the two contributing rivers, unless the contribution from the increase in catchment area will be more significant. Using the lateral discharge option, the surplus water is withdrawn from the model, preserving a proper representation of flood scenarios.

The final aspect are the model parameters. In eq. 2, for instance, apart from the network dimensions and discharge values, gravity acceleration and roughness coefficient have to be defined before the simulation. Another important factor in the simulation, the time step, is defined dynamically by the model. The roughness coefficient was chosen through a relatively simple calibration process. Other large-scale studies did not perform any calibration due to the lack of comparison material with sufficient spatial coverage. Here, the calibration was done comparing our flood map for the historical scenario, prepared as described in section 2.4, with the JRC map (Alfieri et al. 2014). Even though the JRC map was uncalibrated and by necessity only selectively validated, it used more advanced modelling steps that , most likely, have resulted in higher accuracy. The roughness coefficient was assumed to be a constant value throughout each of the seven sub-simulations. In five of them, the best results were achieved with Manning coefficient in the range of 0.13–0.15 s/m$^{1/3}$. Two remaining regions (both in northern Europe) had lower values, likely due to large lake cover. The methodology of map comparison is explained in section 2.5.

## 2.4 Flood extent calculation

Water levels obtained from the model were post-processed by firstly linearly interpolating them along the rivers to increase the density of estimates. For each point, located on average 250 m away from the next point, the nearest





neighbourhood was defined with Thiessen polygons. For each polygon, a constant water level was assumed, therefore extrapolating the water levels over all terrain. Coastal segments were included in the nearest-neighbour calculation in order to avoid a situation where the water levels in a river are extrapolated along the coastline. Elevation from the DEM was then subtracted, per grid cell, from those water levels. From the whole area laying below water levels of the river, only those zones hydrologically connected with the rivers were included. This assumption is presented in Fig. 5: any high terrain protects low-lying terrain behind it.

Similarly to the water level modelling approach there are two main drawbacks. One is the lack of flood volume control, which has large influence on the actual flood extent during an extreme event (Apel et al. 2009). Second, it assumes that anything elevated above the water levels prevents inundation, which neglects the possibility of flood defence failure. However, flood defences can hardly be represented within the resolution of the model. Yet, due to high significance of this aspect, two sets of maps were produced. The first one uses directly the results of the analysis and can therefore be dubbed 'without flood protection' scenario. The second group are then the maps 'with flood protection'. To obtain them, flood defences were assumed to have the same protection standard as calculated by Scussolini et al. (2016) in the FLOPROS database. Further it was assumed that the return periods in those protection standards were equal to return periods of discharges calculated with the Bayesian Network-based model ($Q_p$ in Fig. 5). If extreme discharge is higher than the protection standard ($Q_e > Q_p$), the terrain floods.

Additionally, using the results of Paprotny and Morales Nápoles (2016a) it was possible to calculate, for each climate scenario and river segment, how the return period of discharge would change in the future. This would then indicate if the protection standard will be sufficient in the future. For example, if we consider the 100-year river discharge, a dike with a 200-year protection standard is assumed to prevent a flood of this magnitude in the 1971–2000 time frame. However, if the river discharge increases by 2021–2050 so much that the 100-year discharge calculated from 2021–2050 data equals a 250-year discharge calculated from 1971–2000 data, then $Q_e$ is considered to have a return period of 250 years. Therefore, in this example $Q_e > Q_p$, and on a100-year flood map for 2021–2050 the terrain is marked as inundated.

## 2.5 Reference flood maps

The results of this study ("TUD map") were compared with six "reference" maps: one pan-European map and five regional flood maps. Below we briefly summary the main characteristics of those studies, summarized also in Table 1. The extent of local maps is presented in Fig. 6.

The pan-European map is available from the Joint Research Centre (2014) and it is documented in Alfieri et al. (2014). The map was created by firstly running a rainfall-runoff simulation of river discharges based on interpolated climatological data for 1990–2010. Based on those results, 100-year discharges together with a flood wave hydrograph was estimated; this is the only scenario considered. Two-dimensional hydrodynamic model Lisflood was used to derive flood zones. The study utilized SRTM terrain model and therefore does not include flood defences. The rainfall-runoff model was calibrated against river gauge observations, but the flood extent modelling was not calibrated. The resulting map covers 188,300 km of rivers




(with a 500 km$^2$ catchment area threshold) in a domain that is slightly smaller than the one used here; it omits Cyprus, Iceland and those parts of river basins that are located inside the former Soviet Union territory, except basins of Danube, Vistula and Nemunas. The map's resolution is 100 m and it exactly matches the grid used in the TUD map.

The largest of the regional maps is the Environment Agency's (2016) "Risk of Flooding from Rivers and Sea" map of England. This dataset was produced during 2005–2013 utilizing local-scale modelling and takes into consideration the height, type and condition of the flood defences. The resulting maps were validated locally using experts' assessments. They are continuously updated; the version from April 2015 was used here. The dataset's resolution is 50 m and for the use in this study the flood zones inundated directly from the sea were removed. The map was prepared in four thresholds defined by the flood extents corresponding to return periods: below 30 years, 30–100 years, 100–1000 years, above 1000 years. The largest flood zones are observed in the basins of rivers Great Ouse and Trent. Much less hazard is indicated along the biggest rivers, Severn and Thames.

Two maps from Germany were collected, covering the federal states (*Länder*) of Saxony (Sächsisches Landesamt für Umwelt, Landwirtschaft und Geologie 2016) and Saxony-Anhalt (Landesbetrieb für Hochwasserschutz und Wasserwirtschaft Sachsen-Anhalt 2016). Both were prepared by the states' administration in 2015, but they followed certain national regulations. In both cases, the maps take into account the effect of flood defences and include 1 in 100 years flood scenario. The maps are provided in vector format, but their accuracy ought to be similar to a 1:25,000 map (~25 m). Both regions are almost completely within Elbe river's basin and most of the flood zone is along this river. Another map was obtained for the state of Lower Austria (Amt der NÖ Landesregierung 2016). It is provided in vector format for three scenarios: 30, 100 and 300 years flood. Impact of flood defence structures is included in this map, which was produced in 2012 using 2D modelling. Most of the flood zone is connected with the Danube or its tributary, Morava river.

The final map is from the Swiss canton of Bern (Kanton Bern 2016), which is located within the basin of Aare river, a tributary of the Rhine. It was prepared in 1:5,000 scale from 1997 and 2011 multi-hazard assessments and takes into account the effect of flood defences. However, this is a flood risk map, and due to its graphical representation only the 1 in 300 years flood scenario could be extracted from it. Additionally, this map only includes flood zones that incorporate populated areas. A map for the uninhabited zones exists in lower resolution (1:25,000), albeit it does not include information on return periods. Therefore, the risk map for the 300 years scenario was compared with our 1 in 300 years flood overlay, while the combination of all flood zones from indicated in the two Swiss maps was compared with the 1 in 1000 years map.

The local maps required some modifications for the purpose of comparing them with the pan-European map. They were resampled to 100 m resolution and flood zones related to rivers with catchment areas below 100 km$^2$ (for comparison with the TUD map) and 500 km$^2$ (for comparison with the JRC map) were removed. The latter point was problematic in the sense that flood zones could be connected to multiple rivers, some of which could be below or above the 100/500 km$^2$ threshold; flooding from a larger river can also spread on smaller tributaries. Therefore, similarly to Alfieri et al. (2014), a 1.5 km buffer around the rivers bigger than the threshold was used for selecting flood zones from the full map.



The pan-European map was evaluated with two measures, the same as used by Bates and De Roo (2000) and several later studies. Test for "correctness" (or, "hit rate") indicates what percentage of the reference map is recreated in the pan-European map (eq. 3). As this test does not penalize overestimation, the test for "fit" (or, "Critical Success Index") is applied (eq. 4). They are calculated as follows:

$$I_{cor} = \frac{A_{EM} \cap A_{RM}}{A_{RM}} \times 100 \tag{3}$$

$$I_{fit} = \frac{A_{EM} \cap A_{RM}}{A_{EM} \cup A_{RM}} \times 100 \tag{4}$$

where $A_{EM}$ is the area indicated as flooded in the TUD pan-European map and $A_{RM}$ is the area indicated as flooded in the reference map. The TUD map was compared, using the 100 km$^2$ threshold with the five local maps for all available scenarios, and with the JRC map using the 500 km$^2$ threshold. Both pan-European maps where then compared with five local maps for the 100-year scenario (i.e., without the Swiss map) with a 500 km$^2$ threshold. The results for England and Saxony were split into smaller regions for more detailed overview using Eurostat's (2015) nomenclature of territorial units for statistics (NUTS). England is subdivided into nine statistical regions, while Saxony has three *Direktionsbezirke*, or districts. The comparison between the TUD and JRC map is presented for 7 regions of Europe, the same as the 7 sub-simulations, as in Fig. 4.

## 3 Results

### 3.1 Validation of flood maps

The results of the comparison between the TUD map with reference maps are presented in Table 2. Considering only flood zones connected with catchments bigger than 500 km$^2$, 84% of the JRC's flood zone is also present in the TUD map (indicator $I_{cor}$). However, JRC map indicates 246,000 km$^2$ at risk of flooding within the domain of the TUD map, which in turn shows almost 330,000 km$^2$ within the 100-year flood extent. The average "fit" ($I_{fit}$) is 56%, with the lowest values observed in the western and southern parts of Europe, with more overlap observed in northern Europe. The latter is most likely the effect of numerous lakes which constitute a considerable part of flood layers in both maps of that region.

In the second part of Table 2 the TUD map is compared with local reference maps in all available scenarios. A snapshot of the comparison for Trent river basin in central England is presented in Fig. 7. Large variability in the results is observed; most of the time 50–70% flood zones from the detailed maps are recreated in the TUD maps. Highest value of $I_{cor}$ (up to 78%) was observed in Saxony-Anhalt and some parts of England, and the lowest in Switzerland and parts of Saxony (down to 30%). $I_{cor}$ decreases both in Austria and England between 30-year and 100-year scenarios, though improves again for more extreme floods. $I_{fit}$ is mostly below 30%, but improves when moving from less extreme to more severe scenarios. All local maps include effects of flood defences, therefore this exact pattern would be expected: flood zones expand rapidly with the increase of the return period of flood, as a declining number of defences can withstand the rising water levels. Hence,



variation of the values of $I_{fit}$ can be mostly explained by the differences in flood protection standards. In England, flood defences are mostly expected to protect against return periods of floods of about 75–200 years (Chatterton et al. 2010). Hence, the protection structures shouldn't influence the size of the 1000-year flood zone in England. Indeed, in this scenario and region the highest $I_{cor}$ and $I_{fit}$ values were observed: 69% and 53%, respectively. results were achieved in terms of alignment with the TUD map. Average value of $I_{fit}$ is two times higher (53%) than in the 30-year scenario (25%). Furthermore, the highest protection level in England is expected in London (Scussolini et al. 2016), which had the lowest $I_{fit}$ in the 30-year and 100-year scenario.

In other analysed regions, the flood protection standards are mostly higher, in terms of return periods, than in England: 100–500 years in Germany, 100–1000 years in Austria (highest along the Danube) and 30–200 years in Switzerland (Scussolini et al. 2016, te Linde et al. 2011). For the 100-year scenario, $I_{fit}$ is only 23–27%. In Saxony, Dresden district had lower fit than the other two districts, which is consistent with the fact that the city of Dresden has an improved flood protection level of 500 years as opposed to 100 years in other areas. The test measures used also improve visibly in Austria between 100-year and 300-year scenarios. On the other hand, the lowest performance of the TUD map in Switzerland can be rather explained with the characteristics of the flood map, than high protection standards. The 300-year flood layer could be extracted only for populated areas, which have much better protection than uninhabited areas. The 1000-year flood map is also incomplete, and was compiled for this comparison from flood zones with unknown, but presumably high, return period.

Finally, both pan-European maps are compared with the local reference maps for the 100-year scenario for catchments bigger than 500 km$^2$ (Table 3). In England the performance of the TUD map was better than the JRC's map, though in not all parts of it. When comparing with German and Austrian maps, the performance was similar or slightly lower. Summing up all flooded areas, the results show that the TUD map had higher values of both $I_{cor}$ and $I_{fit}$. Yet, this results could be explained by some drawbacks of the GIS analysis. In particular, it was problematic to filter out the flood zones below the threshold of 500 km$^2$ catchment area. That gives the TUD map a slight advantage over the JRC map, which included only bigger rivers in the simulations. Also, in many areas of England better performance can be attributed to several large zones where both river and coastal floods occur, which favours overestimation of flooded area from the rivers. Lastly, English flood zones are twice as large as the remaining ones taken together. Still, the results of fitting both European maps in Saxony-Anhalt and Lower Austria are very similar. Substantial simplification of the methodology of making the European maps did not result in equal drop in accuracy, but it was largely maintained. The computational time on a regular desktop PC was slightly less than a day per scenario.

## 3.2 Present flood hazard in Europe

River flood hazard maps were prepared and analysed here in two variants: without flood protection and with flood protection as estimated in the FLOPROS database (Scussolini et al. 2016). Full-size images of the maps were included in the supplement. The total area identified within 1000-year flood scenario was almost 389,000 km$^2$, which is about six times



more than the total for coastal flood hazard, if we do not include impact of flood defences. In this section we briefly describe the outcomes of the "historical" scenario (1971–2000).

The majority of the flood zones in the domain were 10-year zones, with only one-sixth belonging to other zones. More than half of the flood hazard was concentrated in only seven countries: Germany, Hungary, France, Romania, Italy, Russia (even though only a small part of this country is included in the domain) and Poland. Splitting the hazard zones by river basin, half of the endangered area is also in only seven of them: Danube (mainly in Austria, Hungary, Serbia and Romania), Neva (Russia), Vistula (Poland), Elbe (mainly Germany), Oder (mostly Poland and Germany), Rhine (mainly Germany) and Po (Italy). 20 river basins with the highest area within flood zones are listed in Fig. 8. Taking into account flood defences, the estimated area of the 1000-year zone is revised downwards only slightly, to 376,000 km$^2$. A decrease in flood extent is noticeable only in Netherlands, where the "dike rings" provide a high level of protection from both coastal and river floods, and Austria, where flood defences along the Danube are considered to have a high protection standard. On the other end of the scale, the 10-year flood zone is mostly constrained to the Dniester river catchment (6400 km$^2$), while the 30-year zone is mostly present in the Balkans and former Soviet Union (basins of Danube, Nemunas, Evros and others).

The country with the largest hazard level proportional to its area is Hungary, as 37% of the country lies within the 1000-year zone. The Netherlands comes second when flood defences are not considered, with 26% of the territory in the flood zone. This value, however, drops to 1% when considering flood protection. Other countries with high fraction of territory in the flood zone include Serbia (24%), Croatia (20%) or Slovakia (14%), all of which are located in the Danube basin. This river system is has not only the biggest basin in the domain and the largest flood extent, but also the highest proportion of flood area compared to total area (15%) among large river basins. Elevated hazard is also present in Po river basin (12%), Weser (10%) and Oder (9%). On the other hand, Nordic countries have low levels of relative hazard, from 1% in Norway to 4% of Finland. Only 3% of the territory is in the hazard zones in Ireland, Portugal, Spain and Switzerland, while in France, the United Kingdom and Austria the figure is 5%, in Poland 8% and in Germany 10%.

## 3.3 Future flood hazard in Europe

The overall size of the river flood hazard zones in Europe increased for all four climate change scenarios considered. Yet, without considering flood defences the increases are small. By mid-century (2021–2050), RCP 4.5 scenario adds 6,500 km$^2$ to the 1000-year zone, while RCP 8.5 adds 8,000 km$^2$ compared to 1971–2000. For 2071–2100, these figures are 17,100 km$^2$ and 9800 km$^2$, respectively (Fig. 9). This constitutes only 1.7–4.4% of the 1971–2000 flood zone. This is largely a result of only modest (on average) increase in river discharge in Europe. As a whole, this corresponds to 5–8% depending on scenario, according to results from Paprotny and Morales Nápoles (2016a). However, the significant implications of changes in discharge becomes apparent when taking into account flood protection standards. The 10-year zone, estimated at 6400 km$^2$ in 1971–2000, is projected to reach 28,000–50,000 km$^2$, depending on time period and emission scenario. The largest expansion in absolute terms was calculated for the 30-year zone, from 43,200 km$^2$ in the end of the 20th century to 130,000–183,000 km$^2$. The 100-year will be larger by around a third (from 215,000 to 275,000–297,000 km$^2$). Smaller changes are





expected in flood hazard with lower probability of occurrence: the 300-year zone is actually projected decrease by 1700–15,700 km$^2$, while the 1000-year zone could add 6700–18,600 km$^2$.

Nevertheless, trends in river flood hazard will be very diversified across Europe. Changes in flood extents presented in Fig. 10 were aggregated to a 50x50 km grid for the sake of clarity. It includes only one scenario (100-year flood), but the

5 trends shown are representative also for other return periods. Fig. 11 shows the relative contributions of each country to the overall change in flood zone size. With or without flood defences, the largest increases in flood hazard area can be observed in central Europe, particularly in Germany, Hungary and Poland. Trends in the Danube basins will be the main source of increase in hazard. Elbe basin will contribute more than the Rhine, while in Poland flood zones along the Oder are projected to expand more than those along Vistula river. Increases could be observed also in France. In the United Kingdom, increases

are observed when flood defences are included, but slight decrease is predicted without taking them into account. Decreases are mostly observed in northern Europe, particularly in Scandinavia, as a large decline of snowfall and, consequently, snowmelt is expected. To a lesser extent, a decrease of flood hazard is projected in many locations around the Mediterranean Sea, which are projected to become drier in the future.

## 4 Discussion

The results have shown that relatively simpler methods can give similar accuracy to more computationally-demanding models for large-scale flood mapping. Three main simplifications were used: river discharges derived from a statistical model; river flow calculated using a one-dimensional 'steady-state' model without channel geometry; flood zones derived in GIS based on water levels from the hydrodynamic model. The similarity in results to the more complex model used by JRC can be traced to the input datasets, which are mostly the same in various flood studies. For example, the SRTM-derived

digital elevation models provides neither the river bed geometry nor the dimensions of flood protection structures. The former can only be obtained through local surveys, despite efforts to approximate river width or depth from global data (Yamazaki et al. 2014). Flood defences were incorporated here using nominal protection standards defined as flood return periods (from Scussolini et al. 2016), but this is only a rough approximation. Yet, as indicated e.g. in Fig. 9, the difference between 'without flood defences' and 'with flood defences' scenarios is immense. Therefore, both present and future flood

hazard and risk estimates need to take this aspect into account. More aspects are related to this issue, such as the influence of flood defences on river flow. Dams retain water from flood waves, while dikes constrain the river to a narrow space between them. Additionally, overtopping is just one of many dike failure mechanisms (Vrijling 2001), while other flood control techniques exist such as bypass channels, e.g. the New Danube that protects Vienna (Kryžanowski et al. 2014). All those analyses are currently feasible only at local or at most national scale, like the recent flood risk assessment in the Netherlands

(Vergouwe 2014). At the European or global level, other techniques will have to be used, such as a formal statistical analysis of the differences between high- and low-resolution maps in order to derive indirect factors that determine the flood protection levels at given locations.



More comparison with local maps would also improve calibration of the large-scale models. So far, other studies have left the models uncalibrated, while here a step has been taken by using JRC's—uncalibrated—flood map. Local maps were not readily available for all sub-simulations, even though all EU countries do such studies. Inter-comparison between the numerous global flood studies could also show what modelling approaches are most efficient. For example, Sampson et al.

(2015) achieved better results than Alfieri et al. (2014) despite using a statistical model of river discharges as input. It would be therefore useful to compare the maps from this study with Sampson's data. We were unable, however, to obtain those by the time the work described here has concluded.

Limitations of input data and models of river flow are not the only sources of uncertainty. Not all flood events are included in the study. Only rivers with catchments that have an area of at least 100 km$^2$ were included in the calculation.

This omits very small rivers where dangerous flash floods can occur, especially in hilly or mountainous terrain (Marchi et al. 2010). Flash floods also appear in places where drainage is insufficient, mainly in urban areas (Nirupama and Simonovic 2007). Moreover, we estimate the extreme river discharge based on two main factors causing flood – rainfall and snowmelt, while floods in northern Europe are also caused by ice and frazil blocking the river flow (Benito et al. 2015). In estuaries, flood hazard is influenced by tides and storm surges, because they might occur at the same time as a river flood (Svensson

and Jones 2004, Petroliagkis et al. 2016). Finally, disastrous floods could be caused by dam breaches (Prettenthaler et al. 2010).

Last but not least, we should mention the uncertainty related with future climate projections. Only one climate model was used in the Bayesian Network model for extreme river discharges. Also, as can be noticed from Fig. 9–11 and the description, difference between RCP 4.5 and RCP 8.5 scenarios is sometimes very large. The uncertainty is therefore

significant and unavoidable, as the differences between models and scenarios are considerable, especially concerning precipitation (Rajczak et al. 2013, Kotlarski et al. 2014). Those aspects, however, do not affect the validation results in section 3.1.

## 5 Conclusions

In this study we have investigated the feasibility of creating pan-European flood maps using a simplified modelling

approach. A one-dimensional 'steady-state' hydrodynamic model of river flow was utilized, with flood zones derived using GIS. It can be concluded this approach largely  fulfilled its aims. Firstly, the method has low computational burden— performing a full simulation for Europe takes less than a day on a regular desktop PC, compared to months that would have been necessary using a more advanced model. Secondly, the comparison with reference flood maps has shown that the method has similar accuracy to a JRC's map, which was made by employing 2-D hydraulic models which are significantly

more expensive computationally. Additionally, the river discharge data used in this study originated from a statistical model instead of a rainfall-runoff model commonly used in other modelling approaches.



The results are also an indication that the resolution and completeness of input data have high importance compared to choice of modelling approach. With river bed geometry and flood defences missing, not much is to be gained by switching from a one-dimensional static model to a two-dimensional dynamic model. Moreover, climate change projections show relatively little change in flood hazard until flood protection standards are incorporated. Yet, the reliability of global flood defence data is rather low, and considerable improvements need to be made. This aspect is where large gains in accuracy of continental or global-scale maps could be made. Then, more detailed digital elevation models are needed as well as data on river beds. Uncertainty of river discharge return periods and their future development should be further reduced by more research into statistical models.

## Data availability

This work relied entirely on public data as inputs, which are available from the providers cited in the paper. Results of the work can be downloaded from an online repository (Paprotny and Morales Nápoles 2016b).

## Acknowledgements

This work was supported by European Union's Seventh Framework Programme under "Risk analysis of infrastructure networks in response to extreme weather" (RAIN) project, grant no. 608166, and European Union's Horizon 2020 research and innovation programme under "Bridging the Gap for Innovations in Disaster resilience" (BRIGAID) project, grant no. 700699.

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

30





**Table 1.** Comparison of main modelling techniques and assumptions in the maps considered in this study.

| Aspect | Pan-European map (TUD) | Pan-European map (JRC) | Local reference maps |
|---|---|---|---|
| River discharge model | Bayesian Network for extreme river discharges (statistical model for Europe) | Rainfall-runoff model (Lisflood) | Mostly river gauge observations |
| Flood scenarios | Peak discharge with a return period assumed to follow Gumbel distribution | Flood hydrograph created with a empirical formula with a return period assumed to follow Gumbel distribution | Discharge with a return period; methodology varies between studies |
| River flow modelling | 1D hydrodynamic model (steady-state), no channel geometry | 2D hydrodynamic model (Lisflood-ACC), no channel geometry | 1D, hybrid 1D/2D or 2D hydrodynamic model, depending on importance of a location and study |
| Calibration of river flow | Based on comparison with JRC map | None | Usually calibrated using river gauge observations |
| Flood zone modelling | Planar approach in GIS | 2D hydrodynamic model (Lisflood-ACC) | 1D, hybrid 1D/2D or 2D hydrodynamic model, depending on importance of a location and study; occasionally GIS only for areas of low importance |
| Validation of results (flood extents) | With local reference maps | With local reference maps | Local knowledge and expertise |
| Output resolution | 100 m | 100 m | 5–50 m |
| Flood defences | Included in post-processing of the maps (estimated protection standard) | Not included | Included in the river flow/flood zone modelling (dimensions, type of defences, sometimes their condition as well) |
| Simulation run time on a desktop computer | 1 day per scenario | Computer cluster used (not feasible on a desktop computer) | From few seconds (1D) to few days (2D) |





**Table 2.** Comparison of the TUD pan-European flood map with reference flood maps.

| Region | | Flood map test measures by return period | | | | | | | |
|---|---|---|---|---|---|---|---|---|---|
| | | 30 years | | 100 years | | 300 years | | 1000 years | |
| | | $I_{cor}$ (%) | $I_{fit}$ (%) | $I_{cor}$ (%) | $I_{fit}$ (%) | $I_{cor}$ (%) | $I_{fit}$ (%) | $I_{cor}$ (%) | $I_{fit}$ (%) |
| **Comparison with the JRC map by sub-simulation, catchments >500 km²** | | | | | | | | | |
| **Full domain** | | | | **83.6** | **55.7** | | | | |
| Central Europe | | | | 82.0 | 58.1 | | | | |
| British Isles and Iberian peninsula | | | | 78.7 | 49.7 | | | | |
| Southern Europe | | | | 81.3 | 49.8 | | | | |
| Western Europe | | | | 77.0 | 51.9 | | | | |
| Danube basin | | | | 86.7 | 54.9 | | | | |
| North-eastern Europe | | | | 87.5 | 61.8 | | | | |
| Scandinavia | | | | 87.1 | 64.8 | | | | |
| **Comparison with local flood maps by NUTS regions, catchments >100 km²** | | | | | | | | | |
| **UKC-UKK** | **England** | **63.2** | **24.6** | **62.4** | **45.1** | | | **68.7** | **53.0** |
| UKC | North East | 56.9 | 23.4 | 54.1 | 34.5 | | | 59.6 | 40.4 |
| UKD | North West | 53.1 | 26.6 | 48.9 | 29.7 | | | 54.0 | 41.4 |
| UKE | Yorkshire and the Humber | 73.2 | 20.7 | 59.3 | 36.8 | | | 68.3 | 48.8 |
| UKF | East Midlands | 63.3 | 18.1 | 62.9 | 46.2 | | | 73.7 | 57.9 |
| UKG | West Midlands | 66.4 | 39.0 | 58.1 | 42.9 | | | 65.9 | 47.2 |
| UKH | East of England | 58.9 | 16.4 | 73.9 | 59.0 | | | 78.0 | 63.2 |
| UKI | London | 70.2 | 15.6 | 46.9 | 19.1 | | | 71.5 | 49.3 |
| UKJ | South East | 65.3 | 36.7 | 59.1 | 42.9 | | | 61.3 | 48.9 |
| UKK | South West | 62.7 | 41.6 | 55.8 | 46.3 | | | 58.5 | 47.2 |
| **DED** | **Sachsen [Saxony]** | . | | **50.4** | **26.6** | | | | |
| DED2 | Dresden | | | 44.9 | 21.5 | | | | |
| DED4 | Chemnitz | | | 33.3 | 23.0 | | | | |
| DED5 | Leipzig | | | 61.5 | 33.7 | | | | |
| **DEE** | **Sachsen-Anhalt [Saxony-Anhalt]** | | | **72.2** | **22.6** | | | | |
| **AT12** | **Niederösterreich [Lower Austria]** | **55.1** | **21.8** | **49.6** | **24.3** | **61.9** | **34.3** | | |
| **CH021** | **Bern** | | | | | **35.9** | **14.5** | **30.2** | **16.8** |




**Table 3.** Comparison of the pan-European flood maps with the local reference flood maps. Includes only river with catchment area bigger than 500 km².

| NUTS | Name | $I_{cor}$ (%) JRC | TUD | $I_{fit}$ (%) JRC | TUD |
|---|---|---|---|---|---|
| **UKC-UKK** | **England** | **51.0** | **77.6** | **39.5** | **43.6** |
| UKC | North East | 54.4 | 67.6 | 38.9 | 40.0 |
| UKD | North West | 49.9 | 52.4 | 36.3 | 25.5 |
| UKE | Yorkshire and the Humber | 62.1 | 76.0 | 37.4 | 33.0 |
| UKF | East Midlands | 54.8 | 77.7 | 43.3 | 37.3 |
| UKG | West Midlands | 73.8 | 74.4 | 56.5 | 46.0 |
| UKH | East of England | 41.3 | 87.3 | 36.7 | 63.5 |
| UKI | London | 59.6 | 71.9 | 20.1 | 16.8 |
| UKJ | South East | 55.5 | 68.7 | 41.4 | 39.8 |
| UKK | South West | 38.2 | 71.3 | 33.6 | 44.0 |
| **DED** | **Sachsen [Saxony]** | **57.3** | **61.1** | **35.7** | **29.2** |
| DED2 | Dresden | 50.0 | 48.4 | 25.1 | 26.5 |
| DED4 | Chemnitz | 44.1 | 56.8 | 42.8 | 39.5 |
| DED5 | Leipzig | 71.2 | 68.1 | 23.5 | 20.0 |
| **DEE** | **Sachsen-Anhalt [Saxony-Anhalt]** | **54.2** | **59.6** | **23.9** | **26.2** |
| **AT12** | **Niederösterreich [Lower Austria]** | **68.9** | **73.8** | **25.8** | **23.4** |
| | **All regions** | **56.3** | **75.0** | **32.1** | **34.1** |

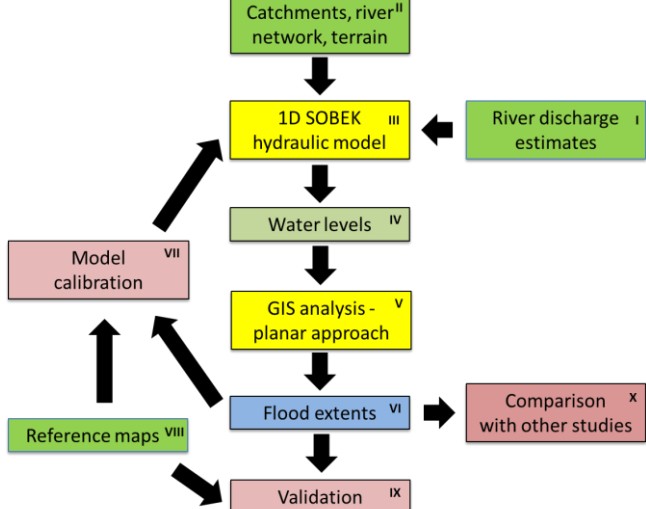

**Figure 1.** Schematic workflow of flood extents calculation. Roman numerals refer to the text.





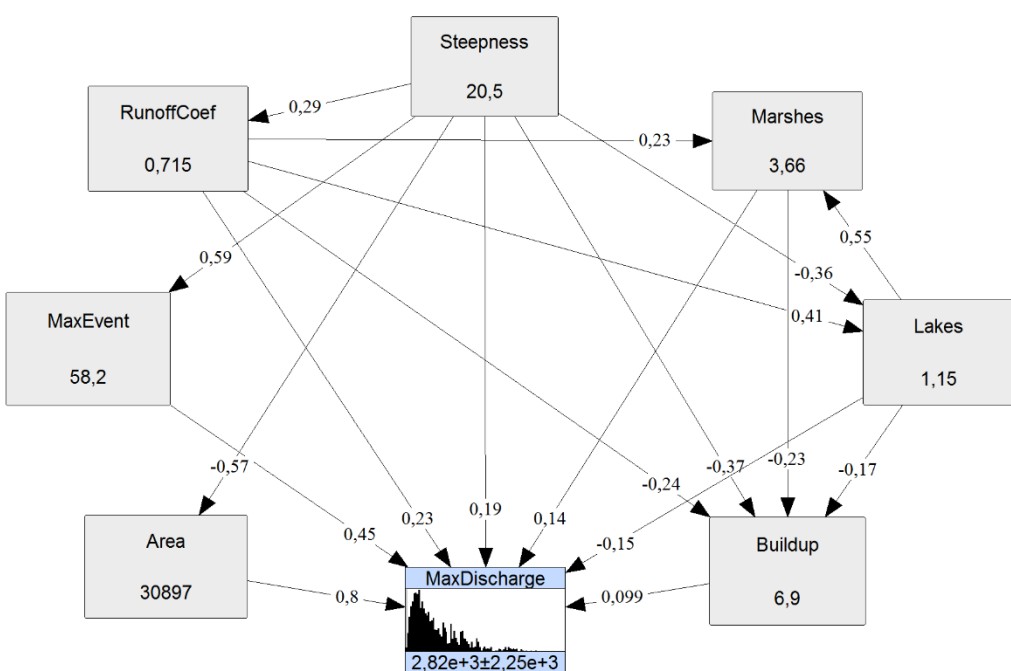

**Figure 2.** Conditionalized Bayesian Network for annual maximum discharge in river Rhine at Basel station in Switzerland in year 2005. The uncertainty distribution of discharge is shown, with a mean of 2820 m³/s ("MaxDischarge").

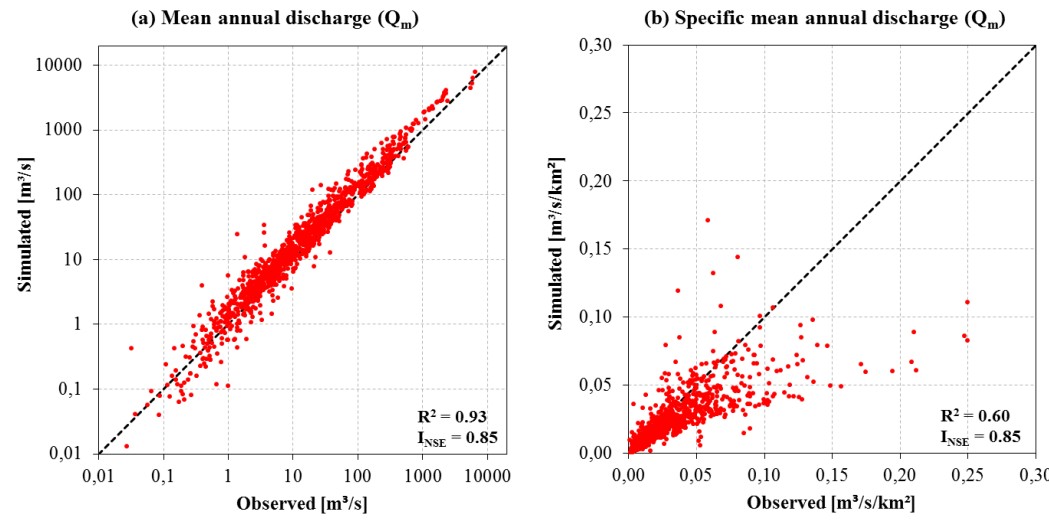

**Figure 3.** Comparison of simulated and observed mean annual river discharges using a Bayesian Network: (a) actual values; (b) specific discharge (runoff divided by the respective catchment area).



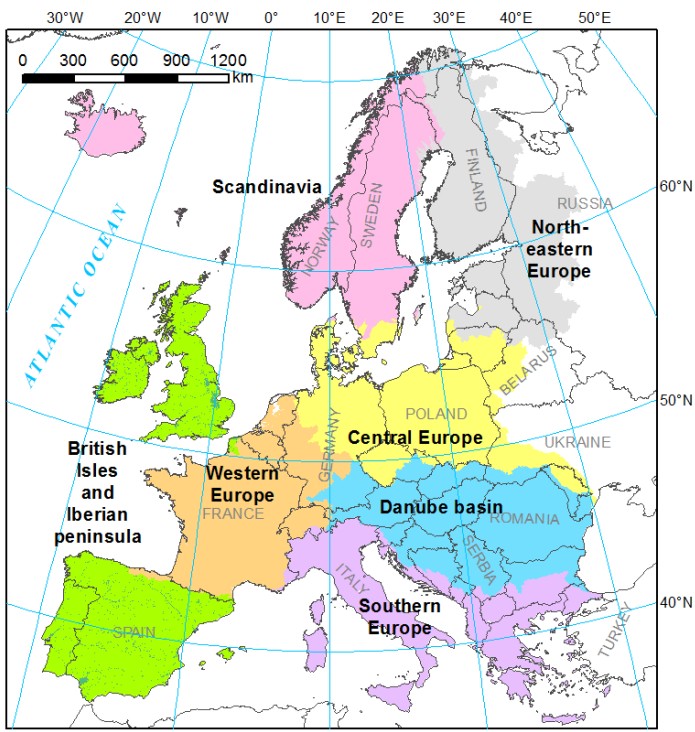

**Figure 4.** Division of the model into 7 sub-simulations, overlaid with political boundaries.

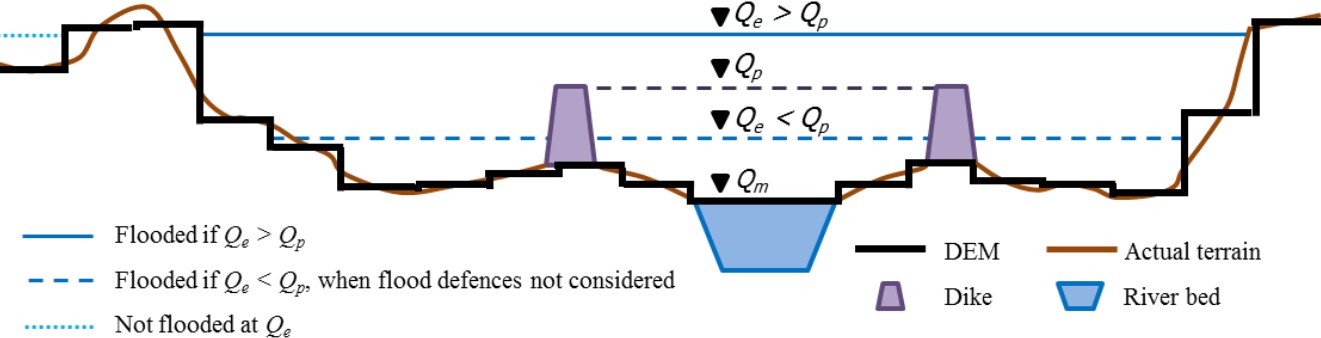

**Figure 5.** Cross-section through a river valley and main model assumptions. The DEM is considered to represent terrain without flood defences and the river's water surface at mean discharge ($Q_m$). Terrain represented in the DEM floods at extreme discharge $Q_e$ if either no flood defences are considered, or when $Q_e > Q_p$, i.e. when extreme discharge is higher than the protection standards. Any higher terrain protects low-lying terrain behind it.





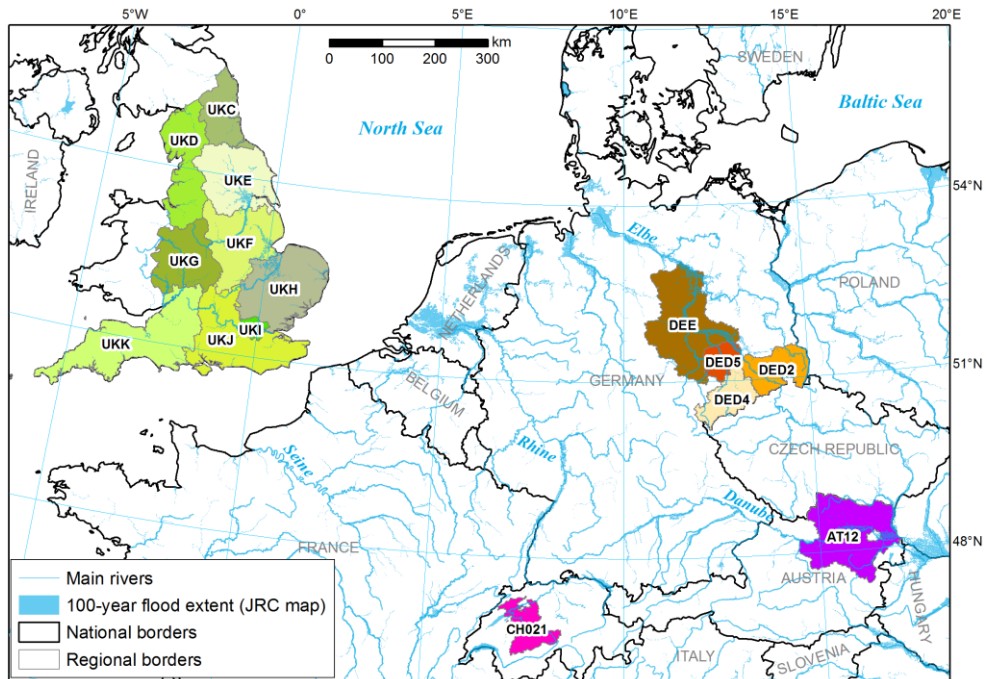

**Figure 6.** Location of the local reference maps with corresponding NUTS codes (see Table 2), with the JRC's flood map (Alfieri et al. 2014) presented in the background.





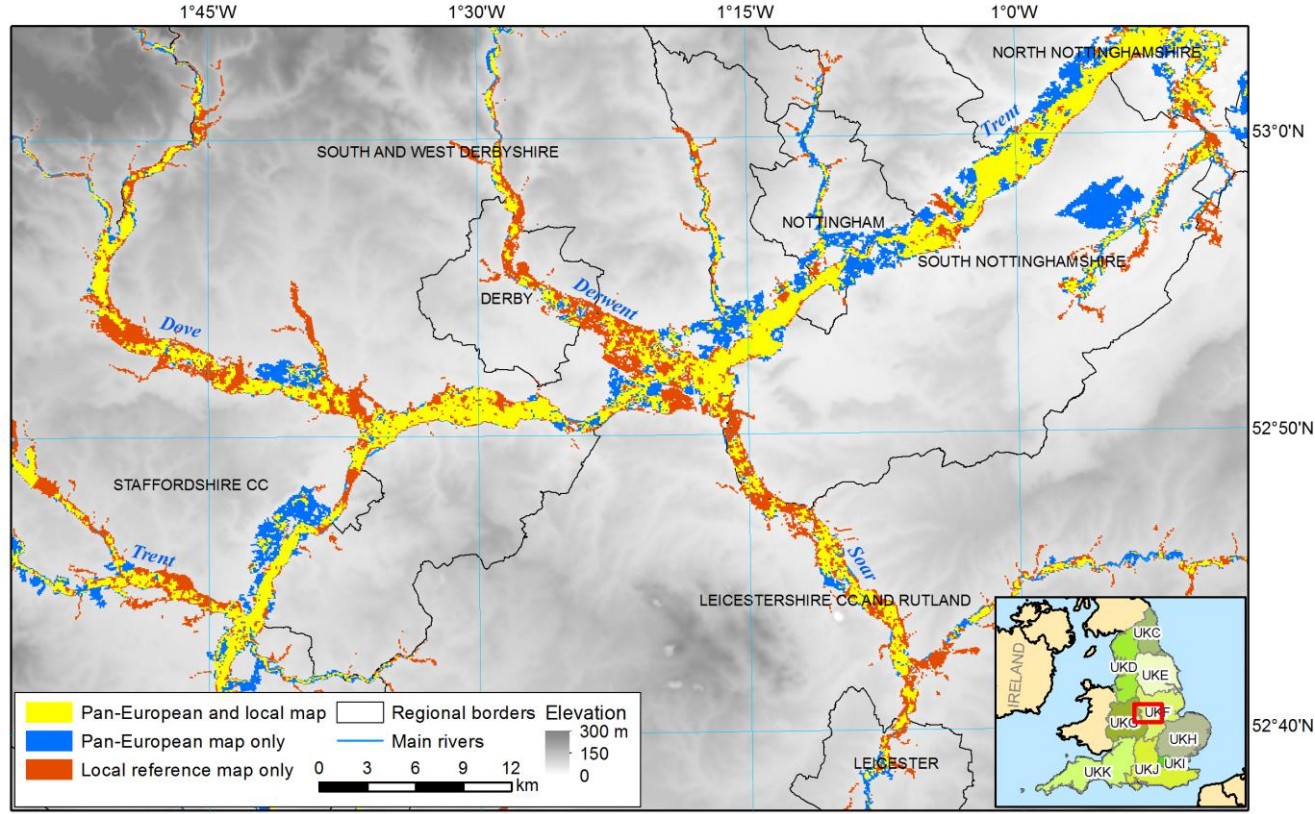

**Figure 7.** An example of the differences between the pan-European map from this study and the local reference map, in this case for the central part of England, both for the 100-year flood scenario (Environment Agency 2016).




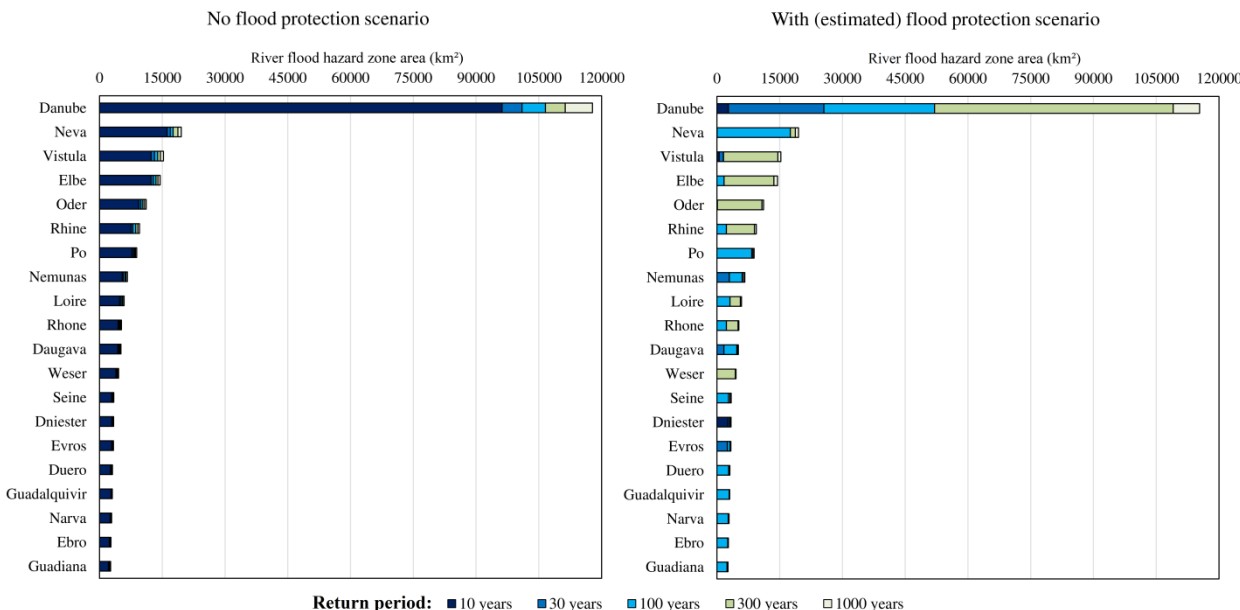

**Figure 8.** Area of flood hazard zones in 20 river basins with the largest hazard, without and with (estimated) flood protection. The basins listed here are highlighted in the maps in the Supplement.

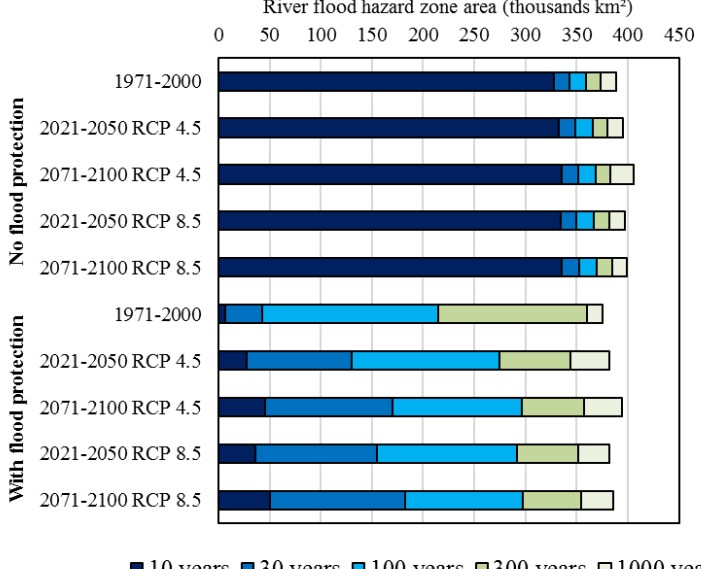

**Figure 9.** Flood hazards zone area in Europe by scenario, without and with (estimated) flood protection. Predictions based on EC-EARTH-COSMO_4.8_clm17 climate model run.



**Figure 10.** Total area of 100-year river flood hazard zones (no flood protection), aggregated to 50x50 km grid, and changes under climate scenarios. Predictions based on EC-EARTH-COSMO_4.8_clm17 climate model run.





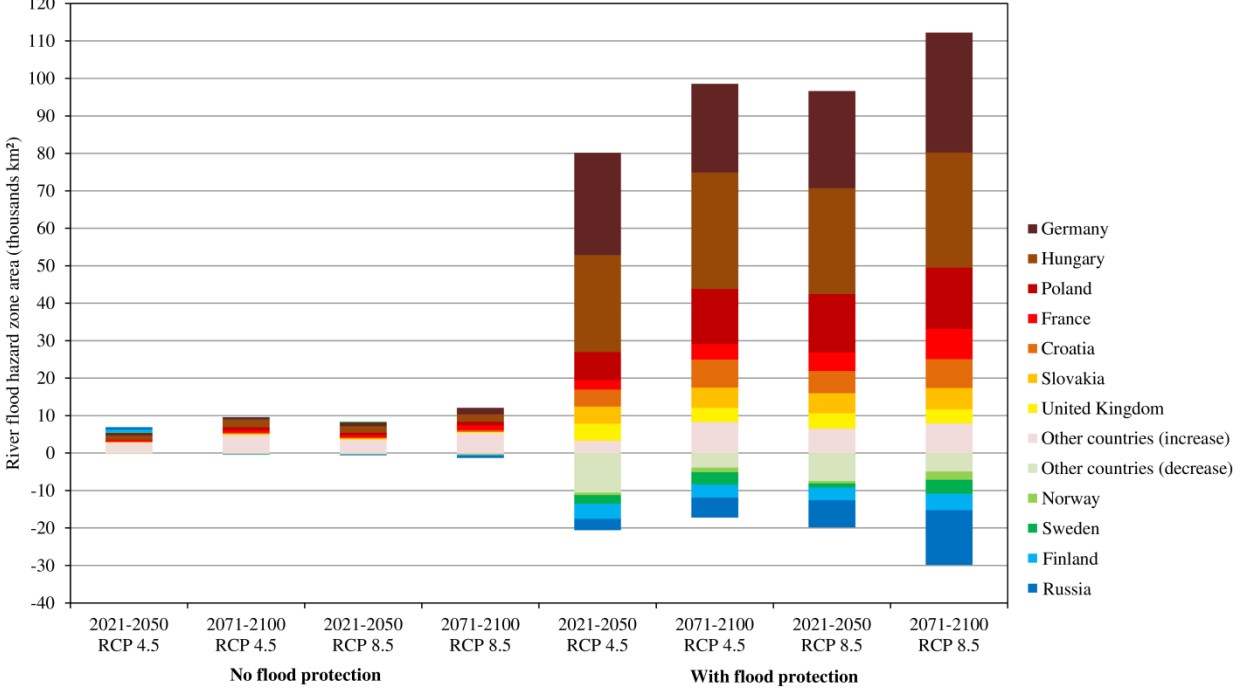

**Figure 11.** Contributions of selected countries to future changes in 100-year flood zone area in Europe by scenario, without and with (estimated) flood protection. Predictions based on EC-EARTH-COSMO_4.8_clm17 climate model run.