# Peer review of "Efficient pan-European river flood hazard modelling through a combination of statistical and physical models"

_Natural Hazards and Earth System Sciences, 2017_

## Referee Comment (RC1) · Anonymous Referee #1 · 19 Jan 2017

The article by Paprotny et al proposes a new simplified methodology to produce flood hazard maps at pan-European level, and is applied to estimate the flood hazard in the current climate and under a few possible climate change scenarios. The overall structure of the article is sensible, though the quality of the writing should be definitely improved. On several occasions I had the feeling that the authors meant something different than what is actually written. In other cases one can guess the meaning but the sentence structure and terminology should be improved. Some examples (WHICH SHOULD BE REPHRASED) include lines 7 of page 1, l 24-25 of p1, l 27-28 p3, l 24-26 p5, l 3-5 p7, l 12-13 p7, l 16-18 p7, l 5-6 p8, l3-4 p13.

I also suggest a review by a native English speaker to correct the typos and punctuation

errors.

Besides what mentioned above, I do not see major reasons for not publishing this article. The proposed approach is relatively novel, though in practice it reflects into using a known method (i.e., 1d hydraulic modeling with planar approximation of water levels), applied to a statistical method used to derive peak discharges (Bayesian Network-based), which was published previously by the authors. As a main comment I suggest the authors to clarify this, including the possible limitations of using a non-volume constraining hydrograph into the hydraulic model (based only on peak discharge), coupled with a non-volume constraining planar approximation of the water levels. Indeed it's clear from the skill scores that results tend to overestimate those of the other regional and JRC maps in terms of flood extent.

Additional Specific comments are listed below

L8 p1: alternative to what? Perhaps novel, or new is more appropriate. Also consider the comments above on the "novelty"

L16 p1: why summarized?

L17 p1: increase of 2-4% compared to what?

I suggest avoiding colloquial forms such as "there is a sharp increase", "This is because", "This fact results in", "in one go", "This fact had", "could be observed"

L 21 p1: I'm surprised to see that the keywords don't include words such as "flood hazard" or "inundation mapping"

L 24, p1: I suggest replacing "common" with costly, damaging, catastrophic or similar

L 26 p1: this is probably because climate scenarios had traditionally high uncertainty to produce hi-res hazard maps. However, some studies did that (e.g., Rojas et al, GEC, 2013)

L 4-5 p2: This sounds unexpected. Is that outside Europe?

L 9 p2: "Alert" should be replaced with "Awareness" in both cases

L 16-17 p2: Also Winsemius et al (NCC, 2015) did a similar validation (see Supplement)

L 27 p2: perhaps "continental" instead of metropolitan?

L31 p2: improve

Sect. 2: I'm surprised not to see some more details on the climatic data used to derive extreme discharges (not even the name of the climatic model!). This is likely to be among the main sources of uncertainties in the output. Some dedicated sentences are utterly needed.

L 8, p4: "affected by flash floods", this needs a further justification, i.e., why flash flood prone catchments cannot be studied with the proposed approach.

L 32 p4: the acronym RCP should be mentioned only after its full name description (now in the following line).

L 1-2 p5: please be careful not to mix the meaning of RCP and that of SSP. In RCP the socio-economic change is not explicitly included.

L 8-9 p5: Why the model is not mentioned here (instead of simply "one of the … regional models")?

L5-6 p6: This sentence is not so informative, unless you state the possible reasons for including these additional terms, and then the reason for neglecting them in your approach.

L 7-8 p7: Not clear what this sentence means. Please clarify. Also, the following sentence should be justified and the parentheses should be removed.

L 20-21 p7: This sounds trivial: I assume gravity acceleration was considered as constant, while the roughness is not specified (perhaps linked to the land cover?). To be clarified. I see it's partly explained later, hence I suggest removing this information here

to avoid confusion.

L4 p8: lying

L 19-23 p8: This part could be improved.

L 27 p9: remove "from"

L20-22: Indeed a more fair evaluation should first remove the permanent water bodies from the flooded area used to assess the skill scores.

L20-22 p 11: meaning areas below 500 kmˆ2 were not excluded in the TUD map? This part is rather confusing. I suggest some clarifications.

L 31-32 p 11: I was not provided with any Supplement, nor can I see it online. I think those files are missing.

L1 p12: how the coastal flood hazard was counted? I understand it's not an outcome of this study.

L18 p 12: remove "has"

L19 p12: "Elevated" should be replaced with a more appropriate alternative.

L25 p 12 onwards. I would find those numbers more intuitive if expressed as percentage of the initial flooded area and/or of the total land area included in the simulation. Absolute numbers are more difficult to evaluate, especially as it refers to a very large domain.

L1 p13: projected to increase

L 10-13 p 13: Those statements require a supporting reference.

L 16 p13: I think that the use of a climate scenario rather than the real climate is an important assumption to mention here, particularly if it is not bias corrected (not mentioned in the article).

L6-7 p 14: I suggest rephrasing this sentence or simply removing as it doesn't seem a good reason.

L26-27 p 14: This sentence sounds rather speculative, as no real comparison is provided on an alternative method. The computing time is clearly a tradeoff between complexity and accuracy pursued. Also, the point on the computing time is not so strong to me, as those maps are just produced ones and don't need frequent updates such as for real time forecasting applications.

L 2-3 p 15: I would be careful here and reformulate this sentence, as you don't give clear evidence to support it.

Table 1: The river flow is calibrated in several stations in the JRC map, as also stated in l 32-33 p8. Or perhaps here (and in the line above) you mean inundation mapping (instead of river flow).

Caption of Fig. 5: I suggest removing the last sentence. In general I don't think it needs such long description.

Caption of Fig. 8: Please provide the Supplement.

Caption of Fig 9-11: What surprises me the most is that only in those 3 captions is the climatic model used as input data mentioned. This needs to be mentioned before in the methods, together with a dedicated description. In addition, given it's only one model, some comments on the uncertainty of the climatic input should be included.

---

## Referee Comment (RC2) · Anonymous Referee #2 · 11 Mar 2017

The manuscript aims to derive a pan-European flood map using the statistical approach of Bayesian Network to derive flood discharge and a 1-D model to propagate flood discharge (using steady state approach) and identify flood prone areas. The same procedural chain is applied in order to capture the effect of climate change on the extension of flooded areas. The proposed method is compared with another pan-European flood map and with different local reference maps produced by different European countries. Surely the authors carried out an important work linked to a EU project and the topic is very interesting. However I have some concerns about the proposed procedure:

1) The authors used a 1D model instead of a 2D model (used in JRC map) – in my opinion the use of 1D model in relatively flat areas like the plains of central Europe

is totally inappropriate. The consequent delineation of flood areas is invalidated by the use of the proposed procedure. Maybe the authors could describe a couple of local study cases (flat areas) in which they compare the use of 1D model against a 2D model.

2) I'm no English mother tongue but some parts of the paper are very hard to read and to understand – I suggest the use of English native speaker to re-read the paper and correct it;

3) I suggest to describe in detail in the paper the structure of Bayesian network used to estimate flood discharge since it is important to understand its structure and its capability if compared to other approaches aimed to the derivation of flood discharge (hydrological models or regionalization procedures).

4) What kind of resampling was used for the reference maps? It is important to point out this aspect since it affects the comparison with the TUD and JRC maps. Also the choice of 1.5 km buffer should be better explained.

5) The authors declare (page 7) that river water level model was calibrated comparing TUD map to JRC map (which was no calibrated). This calibration procedure is a weak point because the authors assume implicitly the need of another existing map to develop their approach. Moreover that introduces in the procedure a sort of bias since the model is forced to reproduce a map, which is not necessarily true. I suggest to not calibrate the model (trying to assess the parameter with ancillary data) and then to carry out the comparison with JRC map and reference maps.

6) In literature other metrics can be used in order to avoid the problems related to the use of Icor. For instance, the authors could use the Kappa index, (Cohen, 1960) which allows a better evaluation of the capability of proposed approach when compared to a reference map.

7) The explanation of the two map sets ("without flood protection" and "with flood protection" – page 8) is very unclear. The authors cite Scussolini et al. 2016 as reference but I think that a clearer explanation is needed to better understand and analyze the results.

8) I'd remove equations (1) and (2) since the DSV equations are well known.

---

## Author Comment (AC1) · 11 Apr 2017

We would like thank the referees for the time spent in reviewing our article and their valuable comments. All the comments and observations will contribute to a significant improvement of the presentation of our study. Below, we list all the comments (C) and our response (R).

Reviewer #1

C: On several occasions I had the feeling that the authors meant something different than what is actually written. In other cases one can guess the meaning but the sentence structure and terminology should be improved. Some examples (WHICH

SHOULD BE REPHRASED) include lines 7 of page 1, l 24-25 of p1, l 27-28 p3, l 24-26 p5, l 3-5 p7, l 12-13 p7, l 16-18 p7, l 5-6 p8, l3-4 p13.

R: The indicated sentences will all be modified to make the meaning more precise and clear.

C: I also suggest a review by a native English speaker to correct the typos and punctuation.

R: The text will be read by a native speaker to make necessary corrections.

C: The proposed approach is relatively novel, though in practice it reflects into using a known method (i.e., 1d hydraulic modeling with planar approximation of water levels), applied to a statistical method used to derive peak discharges (Bayesian Network-based), which was published previously by the authors. As a main comment I suggest the authors to clarify this, including the possible limitations of using a non-volume constraining hydrograph into the hydraulic model (based only on peak discharge), coupled with a non-volume constraining planar approximation of the water levels. Indeed it's clear from the skill scores that results tend to overestimate those of the other regional and JRC maps in terms of flood extent.

R: The reviewer's description summarize well the general approach and results. We will make these points more clear in the abstract, discussion and conclusions, including the fact that the method overestimates flood zones more than the JRC map.

C: L8 p1: alternative to what? Perhaps novel, or new is more appropriate. Also consider the comments above on the "novelty"

R: We mean an alternative to using more complex hydrological models (rainfall-runoff with 2D floodplain modelling). It is indeed not the best expression here, similarly "novel" may not be the best expression. We will rephrase as "In this paper we investigate a different, simplified approach".

C: L16 p1: why summarized?

R: The sentence is indeed unclear, and should have said: "The paper also presents the aggregated results on the flood hazard in Europe, including future projections:.

C: L17 p1: increase of 2-4% compared to what?

R: 2Åň-4% increase by the end of the century compared to the historical scenario (1971-2000). This will be clarified.

C: I suggest avoiding colloquial forms such as "there is a sharp increase", "This is because", "This fact results in", "in one go", "This fact had", "could be observed".

R: The aforementioned expressions will be corrected throughout the manuscript.

C: L 21 p1: I'm surprised to see that the keywords don't include words such as "flood hazard" or "inundation mapping"

R: "Inundation mapping" will be added to keywords, as "flood hazard" is included already in the title.

C: L 24, p1: I suggest replacing "common" with costly, damaging, catastrophic or similar

R: The sentence will be changed to "River floods are one of the most costly natural hazards in Europe"

C: L 26 p1: this is probably because climate scenarios had traditionally high uncertainty to produce hi-res hazard maps. However, some studies did that (e.g., Rojas et al, GEC, 2013)

R: It is true that there are large-scale studies including climate projections in flood hazard, however they are not being applied at local scale. As the reviewer mentions, the uncertainty is high and therefore it is difficult to use the projections in local studies.

C: L 4-5 p2: This sounds unexpected. Is that outside Europe?

R: Yes, we mean that availability of local maps is limited outside the European continent.

C: L 9 p2: "Alert" should be replaced with "Awareness" in both cases

R: That is true, EFAS and GLOFAS are "Awareness" system, not "Alert".

C: L 16-17 p2: Also Winsemius et al (NCC, 2015) did a similar validation (see Supplement)

R: We thank the reviewer for pointing out this publication's supplement. The citation will be added to the paper.

C: L 27 p2: perhaps "continental" instead of metropolitan?

R: "Continental" might be applicable, however "metropolitan" France is the term used in this context, e. g.: https://www.insee.fr/en/statistiques/2382599?sommaire=2382613

C: L31 p2: improve; L4 p8: lying; L 27 p9: remove "from"; L18 p 12: remove "has"

R: The typos will be corrected.

C: Sect. 2: I'm surprised not to see some more details on the climatic data used to derive extreme discharges (not even the name of the climatic model!). This is likely to be among the main sources of uncertainties in the output. Some dedicated sentences are utterly needed. L 8-9 p5: Why the model is not mentioned here (instead of simply "one of the . . . regional models")? Caption of Fig 9-11: What surprises me the most is that only in those 3 captions is the climatic model used as input data mentioned. This needs to be mentioned before in the methods, together with a dedicated description. In addition, given it's only one model, some comments on the uncertainty of the climatic input should be included.

R: Information about the climate data will be added to the methodology: "Here, results from one of the high-resolution (0.11°) regional models operated within EURO-CORDEX framework was used, produced by the Climate Limited-area Modelling-Community utilizing EC-Earth general circulation model (run by ICHEC) with COSMO_4.8_clm17 regional climate model (Rockel et al. 2008), realization r12i1p1

(see Paprotny and Morales Nápoles 2016a for details on datasets used in the European BN model). The comments on uncertainty are included in the discussion section and we will make it clear to the reader in this section.

C: L 8, p4: "affected by flash floods", this needs a further justification, i.e., why flash flood prone catchments cannot be studied with the proposed approach.

R: The smallest rivers are affected by flash floods, for which using daily discharge extremes is not appropriate, as those phenomena last only a few hours or less. The BN model is based on daily discharge, hence it is not desirable to use it for very small catchments. This will be explained in the text.

C: L 32 p4: the acronym RCP should be mentioned only after its full name description (now in the following line).

R: The paragraph will be rewritten to mention the full name first

C: L 1-2 p5: please be careful not to mix the meaning of RCP and that of SSP. In RCP the socio-economic change is not explicitly included.

R: The RCP have underlying socioeconomic assumptions; for clarity the sentence will be rewritten as: "Each of those future scenarios consists of two variants, namely "representative concentration pathways", or RCPs. RCP 4.5 and RCP 8.5 indicate changes in emissions that would cause, by 2100, an increase in radiative forcing by 4.5 or 8.5 W/m2 (Moss et al. 2010)".

C: L5-6 p6: This sentence is not so informative, unless you state the possible reasons for including these additional terms, and then the reason for neglecting them in your approach.

R: Those terms could be used in more detailed simulations using unsteady flow, in situations where they might be of special importance. We will remove the mention of those options from the text.

C: L 7-8 p7: Not clear what this sentence means. Please clarify. Also, the following sentence should be justified and the parentheses should be removed.

R: The sentences will be rewritten: "Meanwhile, the downstream boundaries are the locations were the rivers connect to the sea. The only exceptions are two rivers draining to lake Prespa in the southern Balkans. The boundary was defined as zero water level, representing the mean sea level, unless the DEM indicated a value lower than zero."

C: L 20-21 p7: This sounds trivial: I assume gravity acceleration was considered as constant,

R: That is true, the mention of this will be removed to avoid confusion.

C: while the roughness is not specified (perhaps linked to the land cover?). To be clarified. I see it's partly explained later, hence I suggest removing this information here to avoid confusion.

R: The beginning of the paragraph was streamlined for clarity: "The final aspect to be considered is the model parameters, of which the most important is the roughness coefficient. It was chosen through a relatively simple calibration process." Also as we mention, "The roughness coefficient was assumed to be a constant value throughout each of the seven sub-simulations."

C: L 19-23 p8: This part could be improved.

R: The paragraph will be rewritten to clarify: "For instance, consider a dike that protects against a 200-year flood (Qp), according to FLOPROS. It is therefore sufficient to withstand 100-year river discharge under the historical (1971–2000) scenario. If the extreme river discharges increase due to climate change, the future 100-year event will correspond to river discharge that currently has a return period of more than 100 years, say 250 years. In that case, discharges with a 250-year return period are higher than the 200-year protection standard (Qe > Qp). Therefore, the area that is currently protected against a 100-year event, will be at risk of inundation under climate change"

C: L20-22: Indeed a more fair evaluation should first remove the permanent water bodies from the flooded area used to assess the skill scores.

R: We have recalculated the scores, and we found that while for Scandinavia and North-East Europe the difference in the statistics is substantial, there are negligible change in scores for the reference local maps. We will include that in the revision.

C: L20-22 p 11: meaning areas below 500 kmËĘ2 were not excluded in the TUD map? This part is rather confusing. I suggest some clarifications.

R: No, the areas below 500 km2 were excluded for comparison with the JRC map. However, filtering them out could have been imperfect, as in the original study by Alfieri et al. (2014), therefore slightly increasing overlap between TUD and local maps. We will clarify this as follows: "In particular, it was problematic to completely filter out from the TUD and local maps the flood zones below the threshold of 500 km2 catchment area. That could have increased the overlap between TUD and local maps to a slightly higher degree than the overlap between the JRC and local maps."

C: L 31-32 p 11: I was not provided with any Supplement, nor can I see it online. I think those files are missing; Caption of Fig. 8: Please provide the Supplement.

R: The Supplement was unfortunately missing from the submission due to our omission; this will be fixed for the revision.

C: L1 p12: how the coastal flood hazard was counted? I understand it's not an outcome of this study.

R: The coastal flood extent is taken from the authors' analysis included in Groenemeijer et al. (2016). The missing reference will be added.

C: L19 p12: "Elevated" should be replaced with a more appropriate alternative.

R: "Elevated" will be replaced: "Increased hazard is also present in the Po river basin (12%), Weser (10%) and Oder (9%)."

C: L25 p 12 onwards. I would find those numbers more intuitive if expressed as percentage of the initial flooded area and/or of the total land area included in the simulation. Absolute numbers are more difficult to evaluate, especially as it refers to a very large domain.

R: References to absolute changes will be rewritten as relative changes, e.g. "the 100-year zone is expected to be larger by 28–38%, compared to 215,000 km2 in the historical scenario"

C: L1 p13: projected to increase

R: Not quite, the sentence does contain an error, but the 300-year zone is projected to decrease, not increase, therefore the sentence should read: "…the 300-year zone is actually projected to decrease by 0.7–4.4%, while the 1000-year zone could add 1.8–5%.

C: L 10-13 p 13: Those statements require a supporting reference.

R: The relevant references will be added: Groenemeijer et al. (2016), Rojas et al. (2012)

C: L 16 p13: I think that the use of a climate scenario rather than the real climate is an important assumption to mention here, particularly if it is not bias corrected (not mentioned in the article).

R: It is true that a climate hindcast was used (without bias correction) for the historical scenario, but that aspect was investigated in the previous paper by the authors, detailing the BN model of discharges, where the hindcast gave slightly better results than the reanalysis. R2 of 0.89 compared to 0.79 Given the good overall performance of the model in recreating discharges, shown in this and previous studies, the climate model has little influence on the results presented here (excluding future projections).

C: L6-7 p 14: I suggest rephrasing this sentence or simply removing as it doesn't seem a good reason.

R: The sentence will be removed as it is indeed unnecessary.

C: L26-27 p 14: This sentence sounds rather speculative, as no real comparison is provided on an alternative method. The computing time is clearly a tradeoff between complexity and accuracy pursued. Also, the point on the computing time is not so strong to me, as those maps are just produced ones and don't need frequent updates such as for real time forecasting applications.

R: Indeed the aim was to find a trade-off between complexity and accuracy. Though there is no operational use of the method at the moment, however it can be used to derive flood zones through an ensemble approach much faster and for a larger area than rainfall-runoff and dynamic models. The sentence in question will be modified: "It can be concluded that this approach largely fulfilled its aims of reducing complexity while preserving an acceptable level of accuracy"

C: L 2-3 p 15: I would be careful here and reformulate this sentence, as you don't give clear evidence to support it.

R: The sentence mostly referred to flood protection standards, therefore we think it should be written that: "For instance, the flood protection standards, as modelled in this research, influence the size of the flood zones profoundly, both for the present and future scenarios. The assumption of perfect reliability of flood protection standards could be relaxed and further investigated in future research."

C: Table 1: The river flow is calibrated in several stations in the JRC map, as also stated in l 32-33 p8. Or perhaps here (and in the line above) you mean inundation mapping (instead of river flow).

R: What we meant is that according to Alfieri et al. (2014), in the JRC model discharge is calibrated using observations for part of the domain in the rainfall-runoff model. However, 2D simulation of water levels and flood zones were not calibrated. To be more precise, we will change in the table "River flow modelling" to "Water level modelling".

C: Caption of Fig. 5: I suggest removing the last sentence. In general I don't think it needs such long description.

R: The sentence will be removed from the caption, as it is indeed redundant.

Reviewer #2

C: The authors used a 1D model instead of a 2D model (used in JRC map) – in my opinion the use of 1D model in relatively flat areas like the plains of central Europe is totally inappropriate. The consequent delineation of flood areas is invalidated by the use of the proposed procedure. Maybe the authors could describe a couple of local study cases (flat areas) in which they compare the use of 1D model against a 2D model.

R: Large part of the maps analysed are located in flat areas (England or Elbe and Danube river valleys), and the comparison between 1D and 2D models (albeit with different setups) is an important part of the paper. We also mention previous case studies dealing with 1D and 2D comparison in the introduction.

C: I'm no English mother tongue but some parts of the paper are very hard to read and to understand – I suggest the use of English native speaker to re-read the paper and correct it;

R: The text will be read by a native speaker to make necessary corrections.

C: I suggest to describe in detail in the paper the structure of Bayesian network used to estimate flood discharge since it is important to understand its structure and its capability if compared to other approaches aimed to the derivation of flood discharge (hydrological models or regionalization procedures).

R: The BN model was extensively described and validated in Paprotny and Morales Nápoles (2016a) in Hydrology and Earth System Sciences Discussions, therefore we didn't want to create unnecessary overlaps between the two papers. Yet, we realize that this was not explicitly stated in the manuscript, and will be corrected in the revision,

so it will be clear to the reader were to look for details on the BN model.

C: What kind of resampling was used for the reference maps? It is important to point out this aspect since it affects the comparison with the TUD and JRC maps. Also the choice of 1.5 km buffer should be better explained.

R: The maps were resampled using "cell center" allocation ("the polygon which overlaps the center of the 100 m cell yields the attribute to assign to the cell"). The 1.5 km buffer was chosen purely because it is the same as used in Alfieri et al. (2014), therefore making the comparison more uniform. These observations will be made more clear in the text.

C: The authors declare (page 7) that river water level model was calibrated comparing TUD map to JRC map (which was no calibrated). This calibration procedure is a weak point because the authors assume implicitly the need of another existing map to develop their approach. Moreover that introduces in the procedure a sort of bias since the model is forced to reproduce a map, which is not necessarily true. I suggest to not calibrate the model (trying to assess the parameter with ancillary data) and then to carry out the comparison with JRC map and reference maps.

R: The "calibration" was done primarily to assess how sensitive is the model to change in roughness coefficient. We found that it had minimal influence on the results of comparison presented in the paper. In the table attached to this response (as supplement) we show the statistics for the uncalibrated model (0.15 roughness except for northeastern Europe and Scandinavia, were 0.1 was used) and the results presented in the paper. For 3 out of 7 sub-models, the original value of 0.15 gave best results; those sub-models contained 4 out of 5 local maps. Also, local maps for Saxony-Anhalt and Switzerland were added to our study after all flood maps were obtained from the 1D model. Hence, calibration only influenced results of comparison with maps from Lower Austria, to a negligible degree as can be seen from the table. As the comparison with the local maps is the primary means of validation here, we opted to use the model

results calibrated against the JRC map. Finally, the use of comparison material is not necessary for the present approach to work, as verification of the hydrological model results should always be used whenever possible.

C: In literature other metrics can be used in order to avoid the problems related to the use of Icor. For instance, the authors could use the Kappa index, (Cohen, 1960) which allows a better evaluation of the capability of proposed approach when compared to a reference map.

R: The Kappa index is not applicable in this situation, as it would requires four cases, including a case when both datasets agree that a given area will not be flooded. Moreover, the Kappa index takes values in a range that depends on the way this area is chosen (the whole political territory of the unit analysed, some geographical limits etc.) making it difficult to compare within the study.

C: The explanation of the two map sets ("without flood protection" and "with flood protection" – page 8) is very unclear. The authors cite Scussolini et al. 2016 as reference but I think that a clearer explanation is needed to better understand and analyze the results.

R: The text will be modified: "The second group corresponds to the maps 'with flood protection'. To obtain them, flood defences were assumed to have the same protection standard as calculated by Scussolini et al. (2016) in the FLOPROS database. This dataset provides protection standards defined as return periods of river floods. As a result, it was assumed that the return periods in those protection standards were equal to return periods of discharges calculated with the Bayesian Network-based model (Qp in Fig. 5)." Also per reviewer #1's comment, the second paragraph will be changed more substantially: "For instance, consider a dike that protects against a 200-year flood (Qp), according to FLOPROS. It is therefore sufficient to withstand 100-year river discharge under the historical (1971–2000) scenario. If the extreme river discharges increase due to climate change, the future 100-year event will correspond to river discharge that
currently has a return period of more than 100 years, say 250 years. In that case, discharges with a 250-year return period are higher than the 200-year protection standard (Qe > Qp), therefore area that is currently protected against a 100-year event, will be at risk of inundation under climate change"

C: I'd remove equations (1) and (2) since the DSV equations are well known.

R: Equations (1) and (2) will be removed.

Please also note the supplement to this comment:
http://www.nat-hazards-earth-syst-sci-discuss.net/nhess-2017-4/nhess-2017-4-AC1-supplement.pdf

**Supplement:**

Statistics for the uncalibrated and calibrated model.

| Comparison with JRC model | Uncalibrated | | Calibrated | |
|---|---|---|---|---|
| Sub-model | Correct (%) | Fit (%) | Correct (%) | Fit (%) |
| Central Europe | 82,0% | 58,1% | 82,0% | 58,1% |
| British Isles and Iberian peninsula | 78,7% | 49,7% | 78,7% | 49,7% |
| Southern Europe | 82,6% | 49,7% | 81,3% | 49,8% |
| Western Europe | 77,0% | 51,9% | 77,0% | 51,9% |
| Danube basin | 87,0% | 54,8% | 86,7% | 54,9% |
| North-eastern Europe | 87,4% | 64,8% | 87,1% | 64,8% |
| Scandinavia | 89,6% | 61,1% | 87,5% | 61,8% |
| Comparison with local maps | | | | |
| England | 77,7% | 43,7% | 77,7% | 43,7% |
| Saxony | 61,1% | 29,2% | 61,1% | 29,2% |
| Lower Austria | 60,5% | 26,0% | 59,6% | 26,2% |

---

## Author Response (AR1)

We would like thank the referees and the editor for the time spent in reviewing our article and their valuable comments. All the comments and observations will contribute to a significant improvement of the presentation of our study. Below, we list all the comments and our response, followed by the marked-up version of the manuscript

**Reviewer #1**

- On several occasions I had the feeling that the authors meant something different than what is actually written. In other cases one can guess the meaning but the sentence structure and terminology should be improved. Some examples (WHICH SHOULD BE REPHRASED) include lines 7 of page 1, l 24-25 of p1, l 27-28 p3, l 24-26 p5, l 3-5 p7, l 12-13 p7, l 16-18 p7, l 5-6 p8, l3-4 p13.
  - *The indicated sentences were modified to make the meaning more precise and clear.*
10
- I also suggest a review by a native English speaker to correct the typos and punctuation.
  - *The text was checked by our colleague Antonia Sebastian from the USA, leading to corrections of various language errors throughout.*
- The proposed approach is relatively novel, though in practice it reflects into using a known method (i.e., 1d hydraulic modeling with planar approximation of water levels), applied to a statistical method used to derive peak discharges
15
(Bayesian Network-based), which was published previously by the authors. As a main comment I suggest the authors to clarify this, including the possible limitations of using a non-volume constraining hydrograph into the hydraulic model (based only on peak discharge), coupled with a non-volume constraining planar approximation of the water levels. Indeed it's clear from the skill scores that results tend to overestimate those of the other regional and JRC maps in terms of flood extent.
20
  - *The reviewer's description summarize well the general approach and results. We have made these points more clear in the abstract, discussion and conclusions, including the fact that the method overestimates flood zones more than the JRC map.*
- L8 p1: alternative to what? Perhaps novel, or new is more appropriate. Also consider the comments above on the "novelty"
25
  - *We mean an alternative to using more complex hydrological models (rainfall-runoff with 2D floodplain modelling). It is indeed not the best expression here, similarly "novel" may not be the best expression. We have rephrased it as "In this paper we investigate a different, simplified approach".*
- L16 p1: why summarized?
  - *The sentence is indeed unclear, and should have said: "The paper also presents the aggregated results on*
30
    *the flood hazard in Europe, including future projections".*
- L17 p1: increase of 2-4% compared to what?
  - *2-4% increase by the end of the century compared to the historical scenario (1971-2000). This is clarified now in the text: "We find relatively small changes in flood hazard, i.e. an increase of flood zones area by 2– 4% by the end of the century compared to the historical scenario."*
35
- I suggest avoiding colloquial forms such as "there is a sharp increase", "This is because", "This fact results in", "in one go", "This fact had", "could be observed".
  - *The aforementioned expressions were corrected throughout the manuscript.*
- L 21 p1: I'm surprised to see that the keywords don't include words such as "flood hazard" or "inundation mapping"
  - *"Inundation mapping" was added to keywords, as "flood hazard" is included already in the title.*
40
- L 24, p1: I suggest replacing "common" with costly, damaging, catastrophic or similar
  - *The sentence was changed to "River floods are one of the most costly natural hazards in Europe"*
- L 26 p1: this is probably because climate scenarios had traditionally high uncertainty to produce hi-res hazard maps. However, some studies did that (e.g., Rojas et al, GEC, 2013)
  - *It is true that there are large-scale studies including climate projections in flood hazard, however they are*
45
    *not being applied at local scale. As the reviewer mentions, the uncertainty is high and therefore it is difficult*

*to use the projections in local studies.*

- L 4-5 p2: This sounds unexpected. Is that outside Europe?
  - *Yes, we mean that availability of local maps is limited outside the European continent.*
- L 9 p2: "Alert" should be replaced with "Awareness" in both cases
  - *That is true, EFAS and GLOFAS are "Awareness" systems, not "Alert".*
- L 16-17 p2: Also Winsemius et al (NCC, 2015) did a similar validation (see Supplement)
  - *We thank the reviewer for pointing out this publication's supplement. The citation was added to the paper.*
- L 27 p2: perhaps "continental" instead of metropolitan?
  - *„Continental" might be applicable, however „metropolitan" France is the term used in this context, e. g.: https://www.insee.fr/en/statistiques/2382599?sommaire=2382613*

- L31 p2: improve; L4 p8: lying; L 27 p9: remove "from"; L18 p 12: remove "has"
  - *The typos were corrected.*
- Sect. 2: I'm surprised not to see some more details on the climatic data used to derive extreme discharges (not even the name of the climatic model!). This is likely to be among the main sources of uncertainties in the output. Some dedicated sentences are utterly needed. L 8-9 p5: Why the model is not mentioned here (instead of simply "one of the . . . regional models")? Caption of Fig 9-11: What surprises me the most is that only in those 3 captions is the climatic model used as input data mentioned. This needs to be mentioned before in the methods, together with a dedicated description. In addition, given it's only one model, some comments on the uncertainty of the climatic input should be included.
  - *Information about the climate data was added to the methodology: "Here, results from one of the high-resolution (0.11°) regional models operated within EURO-CORDEX framework was used, produced by the Climate Limited-area Modelling-Community utilizing EC-Earth general circulation model (run by ICHEC) with COSMO_4.8_clm17 regional climate model (Rockel et al. 2008), realization r12i1p1 (see Paprotny and Morales Nápoles 2016a for details on datasets used in the European BN model)." The comments on uncertainty are included in the discussion section and we made it now more clear to the reader in this section.*
- L 8, p4: "affected by flash floods", this needs a further justification, i.e., why flash flood prone catchments cannot be studied with the proposed approach.
  - *The smallest rivers are affected by flash floods, for which using daily discharge extremes is not appropriate, as those phenomena last only a few hours or less. The BN model is based on daily discharge, hence it is not desirable to use it for very small catchments. This is now explained in the text: "Within this domain, the smallest rivers are affected by flash floods and flooding cannot be represented using daily discharge extremes as those phenomena last only a few hours or less"*
- L 32 p4: the acronym RCP should be mentioned only after its full name description (now in the following line).
  - *The paragraph was rewritten to mention the full name first.*
- L 1-2 p5: please be careful not to mix the meaning of RCP and that of SSP. In RCP the socio-economic change is not explicitly included.
  - *The RCPs have underlying socioeconomic assumptions; for clarity the sentence was rewritten as: "Each of those future scenarios consists of two variants, namely "representative concentration pathways" or RCPs. RCP 4.5 and RCP 8.5 indicate changes in emissions that would cause an increase in radiative forcing by 4.5 or 8.5 W/m$^2$ by 2100 (Moss et al. 2010).*
- L5-6 p6: This sentence is not so informative, unless you state the possible reasons for including these additional terms, and then the reason for neglecting them in your approach.
  - *Those terms could be used in more detailed simulations using unsteady flow, in situations where they might be of special importance. We removed the mention of those options from the text.*
- L 7-8 p7: Not clear what this sentence means. Please clarify. Also, the following sentence should be justified and the parentheses should be removed.
  - *The sentences were rewritten as follows: "Meanwhile, the downstream boundaries are the locations were*

*the rivers connect to the sea. The only exceptions are two rivers draining to lake Prespa in the southern Balkans. The boundary was defined as zero water level, representing the mean sea level, unless the DEM indicated a value lower than zero.”*

- L 20-21 p7: This sounds trivial: I assume gravity acceleration was considered as constant,
  - *That is true, the mention of this was removed to avoid confusion.*
- while the roughness is not specified (perhaps linked to the land cover?). To be clarified. I see it's partly explained later, hence I suggest removing this information here to avoid confusion.
  - *The beginning of the paragraph was streamlined for clarity: “The final aspect to be considered is the model parameters, of which the most important is the roughness coefficient. It was chosen through a relatively simple calibration process.” Also as we mention, “The roughness coefficient was assumed to be a constant value throughout each of the seven sub-simulations.”*
- L 19-23 p8: This part could be improved.
  - *The paragraph was rewritten to clarify: “For instance, consider a dike that protects against a 200-year flood ($Q_p$), according to FLOPROS. It is therefore sufficient to withstand 100-year river discharge under the historical (1971–2000) scenario. If the extreme river discharges increase due to climate change, the future 100-year event will correspond to river discharge that currently has a return period of more than 100 years, say 250 years. In that case, discharges with a 250-year return period are higher than the 200-year protection standard ($Q_e > Q_p$). Therefore, the area that is currently protected against a 100-year event, will be at risk of inundation under climate change”*
- L20-22: Indeed a more fair evaluation should first remove the permanent water bodies from the flooded area used to assess the skill scores.
  - *We have recalculated all the scores and we found that while for Scandinavia and North-East Europe the difference in the statistics is substantial, there are only small changes in scores for the reference local maps. At the same time we found that previously published results in Table 3 were mislabelled, i.e. results from Chemnitz and Dresden were switched, and also results for Saxony-Anhalt and Lower Austria were switched. They are now correct. We also accordingly modified the results section.*
- L20-22 p 11: meaning areas below 500 km^2 were not excluded in the TUD map? This part is rather confusing. I suggest some clarifications.
  - *No, the areas below 500 km² were excluded for comparison with the JRC map. However, filtering them out could have been imperfect, as in the original study by Alfieri et al. (2014), therefore slightly increasing overlap between TUD and local maps. We now clarify this as follows: “In particular, it was problematic to completely filter out from the TUD and local maps the flood zones below the threshold of 500 km² catchment area. That could have increased the overlap between TUD and local maps to a slightly higher degree than the overlap between the JRC and local maps.”*
- L 31-32 p 11: I was not provided with any Supplement, nor can I see it online. I think those files are missing; Caption of Fig. 8: Please provide the Supplement.
  - *The Supplement was unfortunately missing from the submission due to our omission; this will be fixed for the revision.*
- L1 p12: how the coastal flood hazard was counted? I understand it's not an outcome of this study.
  - *The coastal flood extent is taken from the authors' analysis included in Groenemeijer et al. (2016). The missing reference was added.*
- L19 p12: “Elevated” should be replaced with a more appropriate alternative.
  - *“Elevated” was replaced with “Increased hazard is also present in the Po river basin (12%), Weser (10%) and Oder (9%).”*
- L25 p 12 onwards. I would find those numbers more intuitive if expressed as percentage of the initial flooded area and/or of the total land area included in the simulation. Absolute numbers are more difficult to evaluate, especially as it refers to a very large domain.
  - *References to absolute changes were rewritten as relative changes, e.g. “the 100-year zone is expected to be larger by 28–38%, compared to 215,000 km² in the historical scenario”*
- L1 p13: projected to increase

- *Not quite, the sentence does contain an error, but the 300-year zone is projected to decrease, not increase, therefore the sentence should have read: "...the 300-year zone is actually projected to decrease by 0.7–4.4%, while the 1000-year zone could add 1.8–5%.*
- L 10-13 p 13: Those statements require a supporting reference.
  - *The relevant references were added: Groenemeijer et al. (2016) and Rojas et al. (2012)*
- L 16 p13: I think that the use of a climate scenario rather than the real climate is an important assumption to mention here, particularly if it is not bias corrected (not mentioned in the article).
  - *It is true that a climate hindcast was used (without bias correction) for the historical scenario, but that aspect was investigated in the previous paper by the authors, detailing the BN model of discharges, where the hindcast gave slightly better results than the reanalysis: $R^2$ of 0.89 compared to 0.79. Given the good overall performance of the model in recreating discharges, shown in this and previous studies, the climate model has little influence on the results presented here (excluding future projections).*
- L6-7 p 14: I suggest rephrasing this sentence or simply removing as it doesn't seem a good reason.
  - *The sentence was removed as it is indeed unnecessary.*
- L26-27 p 14: This sentence sounds rather speculative, as no real comparison is provided on an alternative method. The computing time is clearly a tradeoff between complexity and accuracy pursued. Also, the point on the computing time is not so strong to me, as those maps are just produced ones and don't need frequent updates such as for real time forecasting applications.
  - *Indeed the aim was to find a trade-off between complexity and accuracy. Though there is no operational use of the method at the moment, however it can be used to derive flood zones through an ensemble approach much faster and for a larger area than rainfall-runoff and dynamic models. The sentence in question was modified as follows: "It can be concluded that this approach largely fulfilled its aims of reducing complexity while preserving an acceptable level of accuracy"*
- L 2-3 p 15: I would be careful here and reformulate this sentence, as you don't give clear evidence to support it.
  - *The sentence mostly referred to flood protection standards, therefore we think it should be written that: "For instance, the flood protection standards, as modelled in this research, influence the size of the flood zones profoundly, both for the present and future scenarios. The assumption of perfect reliability of flood protection standards could be relaxed and further investigated in future research."*
- Table 1: The river flow is calibrated in several stations in the JRC map, as also stated in l 32-33 p8. Or perhaps here (and in the line above) you mean inundation mapping (instead of river flow).
  - *What we meant is that according to Alfieri et al. (2014), in the JRC model discharge is calibrated using observations for part of the domain in the rainfall-runoff model. However, 2D simulation of water levels and flood zones were not calibrated. To be more precise, changed in the table "River flow modelling" to "Water level modelling".*
- Caption of Fig. 5: I suggest removing the last sentence. In general I don't think it needs such long description.
  - *The sentence was removed from the caption, as it is indeed redundant.*

**Reviewer #2**

- The authors used a 1D model instead of a 2D model (used in JRC map) – in my opinion the use of 1D model in relatively flat areas like the plains of central Europe is totally inappropriate. The consequent delineation of flood areas is invalidated by the use of the proposed procedure. Maybe the authors could describe a couple of local study cases (flat areas) in which they compare the use of 1D model against a 2D model.
  - *Large part of the maps analysed are located in flat areas (England or Elbe and Danube river valleys), and the comparison between 1D and 2D models (albeit with different setups) is an important part of the paper. We also mention previous case studies dealing with 1D and 2D comparison in the introduction.*
- I'm no English mother tongue but some parts of the paper are very hard to read and to understand – I suggest the use of English native speaker to re-read the paper and correct it;
  - *The text was checked by our colleague Antonia Sebastian from the USA, who provided us with several*

*suggestions for improving the writing, which we have implemented throughout the manuscript.*

- I suggest to describe in detail in the paper the structure of Bayesian network used to estimate flood discharge since it is important to understand its structure and its capability if compared to other approaches aimed to the derivation of flood discharge (hydrological models or regionalization procedures).
    - *The BN model was extensively described and validated in Paprotny and Morales Nápoles (2016a) in Hydrology and Earth System Sciences Discussions, therefore we didn't want to create unnecessary overlaps between the two papers. Yet, we realize that this was not explicitly stated in the manuscript, and was corrected in the revision, so it will be clear to the reader were to look for details on the BN model: "The BN model was extensively described and validated in Paprotny and Morales Nápoles (2016a)."*
- What kind of resampling was used for the reference maps? It is important to point out this aspect since it affects the comparison with the TUD and JRC maps. Also the choice of 1.5 km buffer should be better explained.
    - *The maps were resampled using "cell center" allocation ("the polygon which overlaps the center of the 100 m cell yields the attribute to assign to the cell"). The 1.5 km buffer was chosen purely because it is the same as used in Alfieri et al. (2014), therefore making the comparison more uniform. These observations were made more clear in the text.*
- The authors declare (page 7) that river water level model was calibrated comparing TUD map to JRC map (which was no calibrated). This calibration procedure is a weak point because the authors assume implicitly the need of another existing map to develop their approach. Moreover that introduces in the procedure a sort of bias since the model is forced to reproduce a map, which is not necessarily true. I suggest to not calibrate the model (trying to assess the parameter with ancillary data) and then to carry out the comparison with JRC map and reference maps.
    - *The "calibration" was done primarily to assess how sensitive is the model to change in roughness coefficient. We found that it had minimal influence on the results of comparison presented in the paper. In the table below we show the statistics for the uncalibrated model (0.15 roughness except for north-eastern Europe and Scandinavia, were 0.1 was used) and the results presented in the paper. For 3 out of 7 sub-models, the original value of 0.15 gave best results; those sub-models contained 4 out of 5 local maps. Also, local maps for Saxony-Anhalt and Switzerland were added to our study after all flood maps were obtained from the 1D model. Hence, calibration only influenced results of comparison with maps from Lower Austria, to a negligible degree as can be seen from the table. As the comparison with the local maps is the primary means of validation here, we opted to use the model results calibrated against the JRC map. Finally, we believe that the use of comparison material is not necessary for the present approach to work. We prioritize verification of the hydrological model results whenever possible.*

| Comparison with JRC model | Uncalibrated | | Calibrated | |
|---|---|---|---|---|
| | Correct | | Correct | |
| Sub-model | (%) | Fit (%) | (%) | Fit (%) |
| Central Europe | 82,0% | 58,1% | 82,0% | 58,1% |
| British Isles and Iberian peninsula | 78,7% | 49,7% | 78,7% | 49,7% |
| Southern Europe | 82,6% | 49,7% | 81,3% | 49,8% |
| Western Europe | 77,0% | 51,9% | 77,0% | 51,9% |
| Danube basin | 87,0% | 54,8% | 86,7% | 54,9% |
| North-eastern Europe | 87,4% | 64,8% | 87,1% | 64,8% |

| | | | | |
|---|---|---|---|---|
| Scandinavia | 89,6% | 61,1% | 87,5% | 61,8% |
| Comparison with local maps | | | | |
| England | 77,7% | 43,7% | 77,7% | 43,7% |
| Saxony | 61,1% | 29,2% | 61,1% | 29,2% |
| Lower Austria | 60,5% | 26,0% | 59,6% | 26,2% |

- In literature other metrics can be used in order to avoid the problems related to the use of Icor. For instance, the authors could use the Kappa index, (Cohen, 1960) which allows a better evaluation of the capability of proposed approach when compared to a reference map.
  - *The Kappa index is not applicable in this situation, as it would requires four cases, including a case when both datasets agree that a given area will not be flooded. Moreover, the Kappa index takes values in a range that depends on the way this area is chosen (the whole political territory of the unit analysed, some geographical limits etc.) making it difficult to compare within the study.*
- The explanation of the two map sets ("without flood protection" and "with flood protection" – page 8) is very unclear. The authors cite Scussolini et al. 2016 as reference but I think that a clearer explanation is needed to better understand and analyze the results.
  - The text was modified: *"The second group corresponds to the maps 'with flood protection'. To obtain them, flood defences were assumed to have the same protection standard as calculated by Scussolini et al. (2016) in the FLOPROS database. This dataset provides protection standards defined as return periods of river floods. As a result, it was assumed that the return periods in those protection standards were equal to return periods of discharges calculated with the Bayesian Network-based model ($Q_p$ in Fig. 5)."*
  - Also per reviewer #1's comment, the second paragraph was changed more substantially: "*For instance, consider a dike that protects against a 200-year flood ($Q_p$), according to FLOPROS. It is therefore sufficient to withstand 100-year river discharge under the historical (1971–2000) scenario. If the extreme river discharges increase due to climate change, the future 100-year event will correspond to river discharge that currently has a return period of more than 100 years, say 250 years. In that case, discharges with a 250-year return period are higher than the 200-year protection standard ($Q_e > Q_p$), therefore area that is currently protected against a 100-year event, will be at risk of inundation under climate change*"
- I'd remove equations (1) and (2) since the DSV equations are well known.
  - *We think that for the sake of the narrative and due to the fact that equations (1) and (2) are actually rarely shown, they should be kept in the text.*

[revised manuscript text omitted]

---

## Author Response (AR2)

We would like thank the referees for the time spent in reviewing our article and their valuable comments. All the comments and observations will contribute to a significant improvement of the presentation of our study. Below, we list all the comments and our response to reviewer #2, as reviewer #1 did not have any further comments. Afterwards, we added the marked-up revised manuscript.

5 **Reviewer #2**

- I suggested describing in detail in the paper the structure of Bayesian network used and to remove equations (1) and (2) since these are well known to every hydrologist and are parts of a commercial model (SOBEK). The authors have left the DSV equations (absolutely not useful for the general reader) and do not add any explanation (even synthetic) about the Bayesian network..

  - o *Equations (1) and (2) were now removed and the surrounding text was adjusted accordingly. A new paragraph was added to section 2.2 to provide more information about the Bayesian Network model. We have also referenced to Pearl's original book introducing Bayesian Networks. We should note that the final paper about the Bayesian Network model has just been published in Hydrology and Earth System Sciences (*http://hydrol-earth-syst-sci.net/21/2615/2017/*), hence we have updated the references in the manuscript.*
15
  *The reader can now find the full description of the BN in that paper.*

- About the model calibration process again my observation is that they cannot calibrate their model by comparing their results to JRC map (which was no calibrated). They are trying to reproduce another map maybe created with a similar model. It is very strange that they use a single value of roughness (or a couple of values) for all the test sites and it is strange that their approach is low sensitive to the choice of a key parameter like roughness.

  - o *The relatively low sensitivity of the model is a result of several factors: steady-state simulation, planar approach to flood zones delimitation and the lack of flood defences in the digital elevation model. Nonetheless, the effect of roughness on the model performance had to be analysed, as opposed to utilizing arbitrarily chosen or model-default parameters. As can be noted in Table 1, the JRC map used much different methodology and due to the use of a 2D dynamic is likely to have resulted in somewhat more accurate results*
25
  *over Europe. Calling the process "calibration" is indeed not exactly appropriate, as it was done in an indirect manner. Hence, we changed the expression throughout the manuscript, so that it would not create confusion for the reader, for instance: "This step could indicate, if necessary, new runs of the SOBEK model to adjust the model's roughness coefficient." (Page 4, 21-22).*

- Honestly I do not understand why they cannot apply K-index ("The Kappa index is not applicable in this situation, as
30
  it would requires four cases, including a case when both datasets agree that a given area will not be flooded"). If they have both dataset, it is very simple to derive the four cases even the case not flooded in both dataset.
  - o *We computed that Kappa index for our results and found that generally it gives similar results to $I_{fit}$ measure,*

*however there is noticeable influence of administrative unit size on the K-index. The larger the unit, the larger is the area outside the flood zones, and the higher the K-index. In contrast, the $I_{fit}$ measure we used is not influenced by the size of the administrative unit. It is also easier to interpret for the reader as a percentage measure in [0,1] range than the dimensionless index in [-∞,1] range.*

[revised manuscript text omitted]